# SDS22 coordinates the assembly of holoenzymes from nascent protein phosphatase-1

Xinyu Cao[1], Madryn Lake[1], Gerd Van der Hoeven[1], Zander Claes [1],
Javier del Pino García[1], Sarah Lemaire [1], Elora C. Greiner[2,3],
Spyridoula Karamanou [4], Aleyde Van Eynde [1], Arminja N. Kettenbach [2,3],
Daniel Natera de Benito [5], Laura Carrera García [5],
Cristina Hernando Davalillo[6], Carlos Ortez[5], Andrés Nascimento [5],
Roser Urreizti [7,8] & Mathieu Bollen [1] ✉

SDS22 forms an inactive complex with nascent protein phosphatase PP1 and Inhibitor-3. SDS22:PP1:Inhibitor-3 is a substrate for the ATPase p97/VCP, which liberates PP1 for binding to canonical regulatory subunits. The exact role of SDS22 in PP1-holoenzyme assembly remains elusive. Here, we show that SDS22 stabilizes nascent PP1. In the absence of SDS22, PP1 is gradually lost, resulting in substrate hyperphosphorylation and a proliferation arrest. Similarly, we identify a female individual with a severe neurodevelopmental disorder bearing an unstable SDS22 mutant, associated with decreased PP1 levels. We furthermore find that SDS22 directly binds to Inhibitor-3 and that this is essential for the stable assembly of SDS22:PP1: Inhibitor-3, the recruitment of p97/VCP, and the extraction of SDS22 during holoenzyme assembly. SDS22 with a disabled Inhibitor-3 binding site co-transfers with PP1 to canonical regulatory subunits, thereby forming non-functional holoenzymes. Our data show that SDS22, through simultaneous interaction with PP1 and Inhibitor-3, integrates the major steps of PP1 holoenzyme assembly.

Protein phosphatase-1 (PP1) is a member of the PPP family of protein Ser/Thr phosphatases[1]. It is expressed in all eukaryotic cells and dephosphorylates a wide array of proteins. The selectivity of PP1 largely stems from its association with one or two regulatory subunits, referred to as "Regulatory-Interactors-of-Protein-Phosphatase-One" (RIPPOs), to form specific PP1 holoenzymes[2,3]. Vertebrates express >200 structurally unrelated RIPPOs, accounting for the huge diversity of PP1 holoenzymes, each with its own subset of substrates and mechanism of regulation. RIPPOs specify the function of associated PP1 by blocking or extending its active site or a substrate-binding groove, by recruiting substrates via dedicated domains, and/or by targeting the phosphatase to a specific subcellular location that contains a subset of substrates. The PP1-binding domain of RIPPOs is usually (largely) structurally disordered and contains one to several short linear motifs (SLiMs) that combine to create a dynamic, high-affinity binding interface for PP1. The most widespread SLiM, present in a large majority of RIPPOs, is the so-called RVxF motif, which binds to a hydrophobic groove on

[1]Laboratory of Biosignaling & Therapeutics, KU Leuven Department of Cellular and Molecular Medicine, University of Leuven, B-3000 Leuven, Belgium. [2]Department of Biochemistry and Cell Biology, Geisel School of Medicine at Dartmouth, Hanover, NH, USA. [3]Dartmouth Cancer Center, Lebanon, NH, USA. [4]Laboratory of Molecular Bacteriology, KU Leuven Department of Microbiology and Immunology, University of Leuven, Leuven, Belgium. [5]Neuromuscular Unit, Department of Neurology, Hospital Sant Joan de Deu, Barcelona, Spain. [6]Department of Genetic Medicine–IPER, Hospital Sant Joan de Deu, Barcelona, Spain. [7]Clinical Biochemistry Department, Institut de Recerca Sant Joan de Deu, Hospital Sant Joan de Deu, Barcelona, Spain. [8]Centro de Investigación Biomédica en Red de Enfermedades Raras (CIBERER), Instituto de Salud Carlos III, Madrid, Spain. ✉e-mail: Mathieu.Bollen@kuleuven.be

PP1 that is remote from the active site and functions as a PP1-anchoring motif.

SDS22 (PPP1R7) and Inhibitor-3 (HCGV/IPP3/PPP1R11/TCTEX5) are the two first-evolved and most widespread RIPPOs, hinting at their key role in the regulation of PP1[4,5]. SDS22 is one of only a few RIPPOs with a structured PP1-binding domain. It largely consists of 12 leucine-rich repeats (LRRs) that adopt a banana-shaped structure to generate a large PP1-interaction interface. In contrast, Inhibitor-3 (I3) has an unstructured PP1-binding domain with an RVxF motif and a unique PP1-binding SLiM (CCC motif) with three consecutive cysteines[6]. Since SDS22 and I3 have non-overlapping PP1-binding sites, they can form a ternary complex with PP1[7,8]. SDS22 and I3 are "non-canonical" RIPPOs in that they are not involved in substrate selection or subcellular targeting of PP1, and do not appear to be a component of functional PP1 holoenzymes[5]. Instead, they selectively bind to newly translated PP1, before its transfer to "canonical" RIPPOs[8]. SDS22 locks nascent PP1 in an inactive conformation that lacks one of two metals that are essential for catalysis[9]. In addition, the CCC motif of I3 occludes the active site, thereby preventing the access of substrates[6]. The inactive SDS22:PP1:I3 complex is a substrate for p97/VCP, an AAA$^+$ ATPase that uses ATP hydrolysis to co-extract I3 and SDS22, thereby freeing PP1 for association with canonical RIPPOs to form functional PP1 holoenzymes[8,10–12]. p97/VCP directly targets I3, which is unfolded in the p97/VCP-hexamer channel, and this somehow results in the co-extraction of SDS22. Mature PP1 holoenzymes are highly dynamic because of continuous competition between canonical, SLiM-based RIPPOs for binding to the limited pool of PP1[2,13]. Hence, the diversity of functional PP1 holoenzymes that accumulate in cells is largely determined by the relative abundance of canonical RIPPOs and their (regulated) affinity for PP1.

Several key questions concerning the biogenesis of PP1 holoenzymes remain unanswered. Why does newly translated PP1 selectively form a ternary complex with SDS22 and I3? What prevents I3 in the SDS22:PP1:I3 complex from being dynamically exchanged for other SLiM-based RIPPOs, like RepoMan and MYPT1, that also have non-overlapping PP1-binding sites with SDS22? Why does the extraction of I3 from SDS22:PP1:I3, in contrast to the dissociation of other PP1:RIPPO complexes, require energy? What is the mechanism underlying the co-extraction of SDS22 and I3? What is the function of SDS22 in the ternary SDS22:PP1:I3 complex and in subsequent steps of PP1-holoenzyme assembly?

Here, using specific cellular and molecular research tools, we show that the binding of SDS22 to nascent PP1 is required to stabilize PP1. We describe an individual with an unstable SDS22 mutant and decreased levels of PP1. We also demonstrate that SDS22 and I3 bind simultaneously to PP1 and to each other, making their recruitment selective and irreversible, and their co-extraction energy-dependent. Finally, we show that p97/VCP is only recruited by SDS22:PP1:I3 and that the co-extraction of I3 and SDS22 is essential to generate functional PP1 holoenzymes. Our data provide key molecular insights into the coordination of the major steps of PP1-holoenzyme assembly by SDS22.

## Results

### SDS22 depletion results in a cell-cycle arrest and a post-mitotic re-attachment failure

To explore the function of SDS22 in mammalian cells, we engineered the human HCT116 colorectal carcinoma-cell line, using a CRISPR/Cas9 approach, for the inducible proteolytic degradation of endogenous SDS22[14]. The designed guide RNA targeted the Cas9 protein to a PAM site at the 5′-end of the stop codon of the SDS22-encoding *PPP1R7* alleles (Supplementary Fig. 1a). Donor DNA constructs with 5′- and 3′-homology arms flanking a miniature Auxin-Inducible Degron (mAID), the fluorescent mClover protein (mClover), and a neomycin- or hygromycin-resistance cassette, were used as templates for homologous recombination-mediated repair of Cas9-induced double-strand breaks to generate modified *PPP1R7* alleles encoding a SDS22-mAID-mClover fusion. The used HCT116 cell line also contains a transgene at the AAVS1 locus that expresses the F-box protein TIR1 from *Oryza sativa* in a doxycycline (Dox)-dependent manner. Addition of Dox to induce the expression of TIR1 and IAA (indol-3-acetic acid, a synthetic analog of auxin) to recruit the endogenous SCF-type E3 ubiquitin ligase to mAID leads to the ubiquitination and proteasomal degradation of SDS22-mAID-mClover in the SDS22-degron cell line (Fig. 1a). The SDS22 fusion was already partially degraded 2 h after Dox/IAA addition and became undetectable after 24–48 h, as shown by both immunoblotting (Fig. 1b) and mClover-fluorescence imaging (Supplementary Fig. 1b). However, Dox/IAA addition did not affect the level of SDS22 in the parental cell line (Supplementary Fig. 1c). A washout of IAA in the degron cell line resulted in the re-appearance of SDS22-mAID-mClover after 48 h (Fig. 1b and Supplementary Fig. 1b), attesting to the reversibility of SDS22 depletion.

The Dox/IAA-induced depletion of SDS22 in the HCT116-degron cell line caused a nearly complete proliferation arrest, as detected by both IncuCyte live-cell analysis (Fig. 1c) and SRB assays (Supplementary Fig. 1d). However, Dox/IAA addition only had a minor effect on the proliferation of the parental cell line, and the addition of Dox or IAA alone did not affect the proliferation of either cell line. Genome-wide RNAi-screens (https://depmap.org/portal/) also identified SDS22 as one of only a few RIPPOs that are important for the proliferation of hundreds of cancer-cell lines (Supplementary Fig. 1e). A flow-cytometry analysis of cells released from a $G_1$/S arrest revealed that SDS22-depleted cells were severely delayed in the ensuing cell-cycle progression (Fig. 1d), accounting for their reduced proliferation (Fig. 1c). Time-lapse video imaging of individual cells released from a $G_1$/S arrest showed that SDS22-degron cells that were still proliferating needed more time for transition from S- to M-phase (Fig. 1e) and for progression through M-phase (Fig. 1f). In the latter experiments, the total duration of the Dox/IAA-treatment was limited to 30 h to preclude a nearly total proliferation arrest, as seen with prolonged treatments (Fig. 1c). Global RNA-sequencing disclosed a similar number (≈600) of up- and downregulated genes in SDS22-depleted cells (Fig. 1g; Supplementary Data 1). However, a gene-ontology (GO) pathway analysis of the differentially expressed genes showed a strong enrichment for genes implicated in cell division and DNA replication/repair (Fig. 1h and Supplementary Fig. 1f), in accordance with the deficient cell-cycle progression of SDS22-depleted cells.

In addition to a proliferation arrest, about 25% of SDS22-depleted cells were floating in the medium after a 48 h culture (Supplementary Fig. 1g). When the culture medium was refreshed, without prior cell splitting, a similar % of cells were floating after 48 h, indicating that adherent cells had lost their attachment and/or did not to re-attach after mitosis. Time-lapse video imaging indeed showed that SDS22-depleted cells often failed to re-attach upon completion of cell division (Fig. 1i) or formed loosely attached, slow-growing cell clumps (Fig. 1j). In conclusion, our data revealed that the removal of SDS22 from HCT116 cells culminates in severe cell-cycle progression defects and a post-mitotic cell re-attachment failure.

### SDS22-depleted cells gradually lose PP1, resulting in substrate hyperphosphorylation

Since SDS22 is an established RIPPO, we subsequently investigated whether the severe phenotype of SDS22 depletion is due to changes in the phosphorylation of PP1 substrates, with a specific focus on substrates that are important for cell division and attachment. PP1 is required for spindle-assembly checkpoint (SAC) silencing during metaphase, once all kinetochores form correct amphitelic microtubule attachments[15,16]. Moreover, in fission yeast, SDS22 is essential for the metaphase–anaphase transition[17], consistent with a role in SAC silencing. To examine whether deficient SAC silencing accounts for the

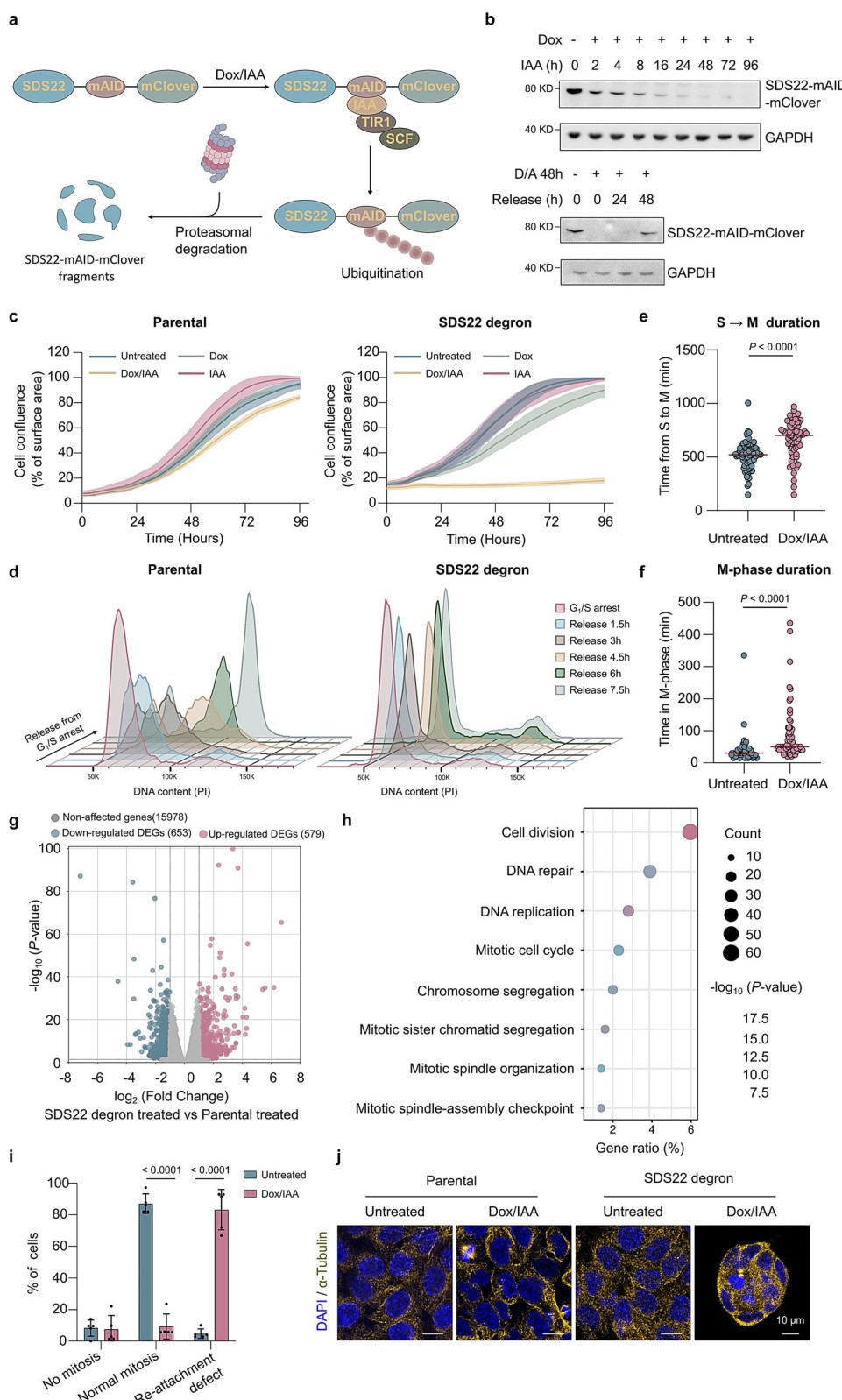

prolonged mitosis in SDS2-depleted cells, we measured the mitotic duration of HeLa cells before and after the knockdown of SDS22 and/or MAD2 (Fig. 2a and Supplementary Fig. 2a). MAD2 is a key component of SAC signaling and its removal results in a SAC override. As in SDS22-depleted HCT116 cells (Fig. 1f), the knockdown of SDS22 in HeLa cells also prolonged mitosis, and this phenotype was completely rescued by the simultaneous knockdown of MAD2 (Fig. 2a and

Supplementary Fig. 2a). This showed that the prolonged mitosis of SDS22-depleted cells stems from a failure to silence the SAC. In accordance with this conclusion, time-lapse imaging revealed that the depletion of SDS22 in HeLa cells resulted in a prolonged metaphase (Supplementary Fig. 2b, c), but had no effect on the prophase → metaphase duration (Supplementary Fig. 2d), arguing against rate-limiting effects of SDS22 depletion on achieving stable

**Fig. 1 | The degradation of SDS22 leads to a proliferation arrest. a** Scheme of Dox/IAA-induced degradation of SDS22-mAID-mClover. **b** Time course and reversibility of SDS22-mAID-mClover degradation. Dox was added 18 h before IAA (upper panel). SDS22-mAID-mClover re-accumulation after Dox/IAA washout (bottom panel). **c** The effect of SDS22 depletion on cell proliferation (IncuCyte). The cells were untreated or pretreated with Dox and/or IAA for 6 h, before the start of scanning. The solid lines are means of three technical replicates and the shaded areas show the SD range from a single data set, representative for three experiments. **d** The effect of SDS22 degradation on cell-cycle progression. Dox/IAA-treated cells were arrested in $G_1/S$, released from thymidine, and analyzed by flow cytometry after propidium iodide (PI) staining. **e** Quantification of S → M duration in SDS22-degron cells, (non-)treated with Dox/IAA for 8 h, by time-lapse imaging of individual cells. S → M duration was defined as the time between release from a single-thymidine arrest and cell rounding. Cells were imaged at 10-min intervals. The $P$ value is from two-sided unpaired t-test (n = 78 cells for each conditions). **f** Cells treated as in (**e**), but scored by time-lapse imaging for the duration of M-phase (time from cell rounding to either cell flattening or formation of loosely attached cell clumps). The $P$ value is from two-sided unpaired t-test (n = 73 cells for each conditions). **g** Volcano plot of RNA-sequencing data (n = 3) showing differentially expressed genes (DEGs) in Dox/IAA-treated (48 h) SDS22-degron versus parental cell lines. The $P$ values of the likelihood ratio test were calculated by edgeR. The cut-off for the DEGs was set at $P$ value < 0.05 and logFC > 1. **h** Dot plot of the gene-ontology (GO) pathway enrichment analysis for the DEGs **i** Mitotic phenotypes in SDS22-depleted degron cells, as quantified by live time-lapse imaging. Details as in (**d**). The data are expressed as means ± SD (n = 5 independent experiments; >50 cells analyzed in each condition). $P$ values were from two-sided unpaired t-test. **j** Morphological changes in parental and SDS22-degron cells, treated or not with Dox/IAA. Cells were fixed and stained for DAPI (blue) and α-tubulin (yellow). Scale bars, 10 μm.

kinetochore–microtubule attachments. The extended mitosis of SDS22-depleted cells was rescued by expression of siRNA-resistant EGFP-tagged wildtype SDS22 (SDS22-WT) (Fig. 2b and Supplementary Fig. 2e). However, previously described PP1-binding mutants of SDS22 (SDS22-M)[18] only partially rescued this phenotype, and the extent of the rescue was inversely correlated with the number (1→4) of mutated PP1-binding residues. These data suggested that the failure of SAC silencing in SDS22-depleted cells was due to a deficient dephosphorylation of SAC proteins by PP1. Consistent with this interpretation, a global phospho-proteome analysis disclosed hyperphosphorylation of key SAC components in SDS22-depleted HCT116 cells, including KNL1, CDC20, BUB1, BUB3, PLK1 and Aurora B (Fig. 2c and Supplementary Fig. 2f; Supplementary Data 2). A GO-analysis of affected pathways in SDS22-depleted cells showed a clear overlap between the global phosphoproteome (Supplementary Fig. 2g) and RNA-sequencing data (Fig. 1g), with a strong enrichment for cell division (Fig. 2d, Supplementary Figs. 1f and 2g).

ERM (Ezrin, Radixin and Moesin) proteins link the plasma membrane to the cortical actin cytoskeleton[19]. Their phosphorylation at the mitotic entry stiffens the cell cortex, resulting in cell rounding, while their dephosphorylation by PP1 at the mitotic exit is required for cell flattening and re-attachment. In the HCT116-degron cell line, ERM proteins became strongly hyperphosphorylated upon depletion of SDS22, but their phosphorylation level normalized after the re-accumulation of SDS22 in the absence of Dox/IAA (Fig. 2e). We also noted a strong staining for phosphorylated ERM proteins (pERM) in SDS22-depleted cells (Fig. 2f and Supplementary Fig. 2h). ERM hyperphosphorylation was also detected in SDS22-depleted cells that were synchronized in prometaphase (nocodazole addition) and subsequently released by nocodazole washout (Supplementary Fig. 2i). To explore whether ERM hyperphosphorylation was caused by deficient ERM dephosphorylation, we performed a cell-based ERM phosphatase assay (Fig. 2g). Dox/IAA-treated parental and degron cells were first treated with the cell-permeable PPP-type phosphatase inhibitor calyculin A to obtain maximally phosphorylated ERM. Subsequently, the cells were resuspended in fresh medium with 50 nM of the kinase inhibitor staurosporine. This resulted in a rapid and complete dephosphorylation of ERM proteins in the parental cells. However, a portion of the ERM proteins in the degron cells remained phosphorylated, hinting at a dephosphorylation deficit. This difference between parental and degron cell lines was not seen, however, in cells that were not treated with Dox/IAA (Supplementary Fig. 2j).

The above data demonstrated that PP1-mediated SAC silencing and ERM dephosphorylation is hampered in SDS22-depleted cells and in cells expressing a PP1-binding mutant of SDS22. As SDS22-associated PP1 is catalytically inactive[7,9], the contribution of SDS22 to SAC silencing and ERM dephosphorylation must be indirect. In view of the role of SDS22 in the maturation of newly translated PP1 (see Introduction), we examined whether SDS22 depletion affects the PP1

protein level. The Dox/IAA-induced depletion of SDS22 indeed resulted in a gradual loss of PP1, as detected by immunoblotting with an antibody that recognizes all PP1 isoforms (Fig. 2h, i). Eventually, 96 h after the induction of SDS22 degradation, the PP1 level was decreased by about 80%. We verified that the loss of PP1 in SDS22-depleted cells was not caused by decreased transcription, as the transcript levels of the PP1α, β and γ isoforms were not affected by SDS22 depletion (Supplementary Fig. 2k). Collectively, our data strongly suggest that the severe phenotype associated with the depletion of SDS22 stems from a loss of PP1 protein, resulting in hyperphosphorylation of PP1 substrates and misregulation of cellular processes that are critically dependent on PP1, including SAC silencing and cell re-attachment after cell division.

## Loss of SDS22 and PP1 in a patient with severe neurodevelopmental delay

A 10-year-old female patient, further referred to as P1, exhibited a psychomotor developmental delay from the early months of life. Additionally, she experienced febrile seizures from 14 months to 3 years of age. Throughout her life, she never achieved the ability to sit, walk independently or speak. The child presents with severe intellectual disability, pronounced hypotonia with present osteotendinous reflexes, hyperkinesia, choreodystonic movements, ptosis and swallowing difficulties. Comprehensive metabolic studies of blood, urine and cerebrospinal fluid did not yield aberrant results (Supplementary Note 1; Supplementary Tables 1 and 2). At the age of 8, brain magnetic resonance imaging (MRI) revealed a reduction in the volume of white matter, nerves, the optic chiasm, thalami and the medulla oblongata, with no signal abnormalities. Electromyography and nerve conduction velocities appeared within the normal range (Supplementary Fig. 3a). However, a muscle biopsy performed at the age of 4 showed minimal, non-specific changes (Supplementary Note 1).

Whole exome sequencing (WES) trio analyses revealed two homozygous variants (in *SACS* and *NID1* genes), with only the mother being heterozygous (Supplementary Data 3). MLPA and microsatellite analyses were consistent with maternal isodisomy of chromosome 13. This was further confirmed by whole-genome SNP array (750 K Cytoscan, Applied Biosystems), showing a female hybridization pattern with no evidence of clinically significant copy number alterations according to current knowledge, and revealing loss of heterozygosity (LOH) in two segments of chromosome 13 spanning 61.6 Mb (arr[GRCh37] 13q11q12.3(19450957_31639909) x2 hmz, 13q14.13q32.1(46699801_96106269)x2 hmz). Detailed analysis of the WES data and subsequent mRNASeq analysis did not disclose any putative pathogenic variants within this region. Therefore, the partial heteroisodisomy is unlikely to be the cause of the severe phenotype of P1.

Importantly, further analysis of the WES data also uncovered a de novo heterozygous mutation (2:242109282G > A; c.906G >

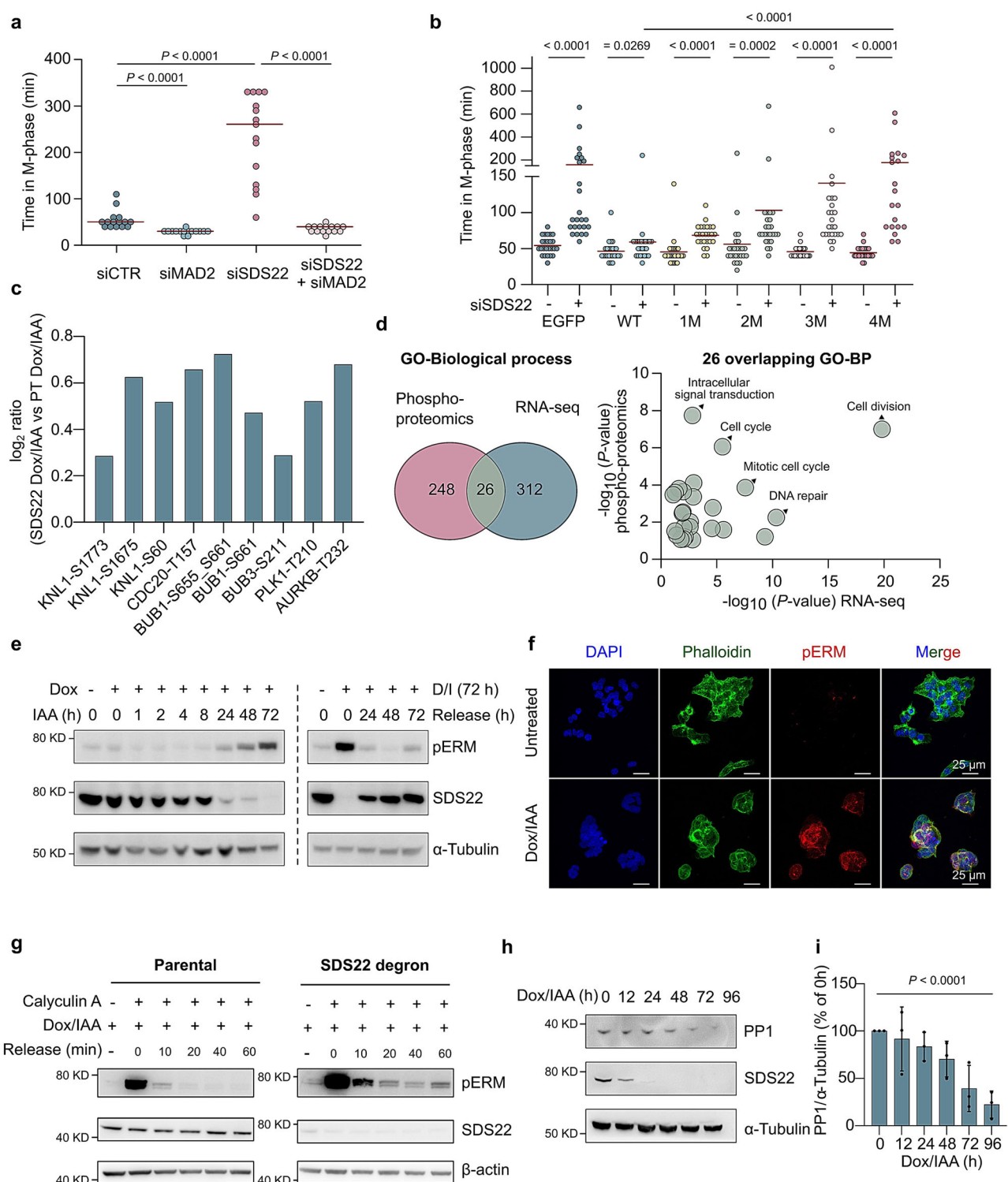

A:p.Trp302*) in the SDS22-encoding *PPP1R7* gene, absent in gnomAD v2 (Supplementary Fig. 3b). This mutation was only observed in 24% of the transcripts (out of a total of 124 reads), which is less than expected for a heterozygous mutation (50%), suggesting that the mutated SDS22 transcript is about twofold less stable than the wildtype transcript. Moreover, the mutation in patient P1 introduces a stop codon in exon 10 of *PPP1R7*[20], putatively resulting in the expression of a C-terminally nicked SDS22 variant, lacking the last 58 residues comprising LRR11, LRR12 and the LRR-Cap structure (Fig. 3a). To verify the consequences of the heterozygous *PPP1R7* c.906G > A mutation on the expression levels of SDS22 and PP1, we isolated fibroblasts from a skin biopsy of P1

and four healthy controls, and found that the level of SDS22 in the cells of patient P1, as compared to that in controls, was decreased by about 80% (Fig. 3b). This decrease was larger than expected from a heterozygous mutation, suggesting that SDS22-(1-301) in P1 may function as a dominant-negative mutant. Importantly, the level of PP1 in the patient's fibroblasts was also decreased by about 65%. Collectively, these data indicated that the mutated SDS22 transcript and protein in patient P1 are unstable and that the resulting decreased expression levels of SDS22 are associated with a co-depletion of PP1. Multiple efforts to identify additional patients on matchmaking platforms with Trp302* mutations in SDS22 were unsuccessful thus far.

**Fig. 2 | Misregulation of PP1 in SDS22-depleted cells. a** Time-lapse imaging (10-min interval) of individual HeLa cells released from a single-thymidine arrest after treatment with siRNAs (siCTR, siMAD2 and/or siMAD2). *P* values are from two-sided Mann–Whitney U test (n = 15 cells for each condition). **b** Rescue of prolonged M-phase in SDS22-depleted HeLa cells by expression of EGFP-tagged SDS22-WT or a PP1-binding mutant of SDS22 (1M: E192A; 2M: E192A + E300A; 3M: F170A + E192A + E300A; 4M: F170A + F214A + E192A + E300A). *P* values are from two-sided Mann–Whitney U test (n = 25 cells/conditions). **c** Phosphopeptides from SAC components in SDS22-degron cells as compared to that in parental cells, treated with Dox/IAA. **d** Venn diagram of the affected biological processes (BP) after SDS22 degradation in the phosphoproteomics and RNA-sequencing data sets, as determined by GO-pathway analysis (left panel). The right panel shows the significance (−log$_{10}$ of the *P* values) of the 26 overlapping pathways. **e** Time-dependent degradation of SDS22 and ERM phosphorylation (pERM). IAA was added for the indicated times, in the presence of Dox, to induce the degradation of SDS22 (left). Also shown is the effect of a subsequent Dox/IAA (D/I) washout for 0 → 72 h on the recovery of SDS22 protein and ERM dephosphorylation (right panel). **f** Immunofluorescence staining of SDS22-degron cells, either untreated or treated with Dox/IAA for 72 h. The fixed cells were stained for DNA (DAPI), phalloidin and pERM. The scale bars are 25 μm. **g** Cell-based assay of ERM dephosphorylation. Parental and SDS22-degron cells were incubated with Dox/IAA and thymidine. Calyculin A (25 nM, 30 min) served to maximize ERM phosphorylation. Subsequently, the cells were released in medium with 50 nM staurosporine to enable ERM dephosphorylation. **h** Time course of SDS22 depletion in the degron cell line and its corresponding effect on PP1 protein in cell lysates. The cell lysates were prepared from the combined attached plus floating cells. **i** Quantification of the data for PP1 shown in (**h**). The bars represent means ± SD (n = 3 independent experiments). The *P* value is from two-sided unpaired t-test.

We have subsequently compared the fate of EGFP-tagged SDS22-WT and SDS22-W302* in transiently transfected HEK293T cells. Traps of EGFP-SDS22-WT contained PP1, I3 and BCLAF1, but these ligands were not detected in traps of EGFP-SDS22-W302* (Fig. 3c), showing that the SDS22 mutant of patient P1 is dysfunctional. Moreover, SDS22-W302* was expressed at much lower levels than EGFP-SDS22-WT (Fig. 3c–e), but this phenotype was largely rescued by the addition of the proteasome inhibitor MG132 (Fig. 3d, e). This suggested that SDS22-W302* was less abundant because of an increased degradation rate. Finally, we noted that SDS22-W302*, before and after treatment with MG132, largely appeared in granules, hinting at an increased tendency to aggregate (Fig. 3f). Collectively, these data validated and extended our observations on SDS22-W302* in fibroblasts of P1, and demonstrated that this mutant is deficient in ligand binding, less soluble and readily targeted for proteasomal degradation, consistent with previous data on the essential role of the C-terminus of SDS22[21,22].

## SDS22 also interacts directly with I3

While the previous data disclosed a key role for SDS22 in stabilizing newly translated PP1, they did not rule out additional functions for SDS22 in the SDS22:PP1:I3 complex, for example, related to I3 recruitment or the later steps of PP1-holoenzyme assembly. Modeling of the ternary complex using the AlphaFold-Multimer tool correctly predicted with high confidence the experimentally determined interaction sites of SDS22 and I3 for PP1 (Fig. 4a, b and Supplementary Fig. 4a–d)[23]. Thus, SDS22 mainly bound to PP1 via its LRR repeats, consistent with crystallographic data[9,18]. I3 has a degenerate RVxF motif that properly docked to its well-characterized hydrophobic binding groove on PP1[6,24]. The model also correctly predicted that the CCC motif of I3 occludes the active site of PP1, as recently demonstrated[6]. Unexpectedly, the AlphaFold-Multimer model of SDS22:PP1:I3 also predicted a hitherto unknown direct interaction between SDS22 and I3. Co-precipitation experiments using purified His-I3 and SDS22 confirmed their direct interaction, independent of PP1 (Fig. 4c). To further validate the SDS22:I3 interaction in a cellular context, we transiently transfected HEK293T cells with EGFP-tagged I3-fragments and examined the interaction of the trapped fusions with PP1 and SDS22 (Fig. 4d, e). This analysis disclosed a key role for I3 residues 71–84 in the interaction with SDS22 which, however, were not required for PP1 binding. I3-(71–84) comprises four acidic residues that were predicted to interact with seven basic residues at the concave side of LRR3-8 of SDS22 (Fig. 4b, f, g). All of these charged residues of I3 and SDS22 are phylogenetically conserved (Fig. 4g and Supplementary Fig. 4e, f). Moreover, alanine mutation of the four implicated acidic residues in EGFP-I3 (I3-DE4A) abolished its interaction with SDS22 in EGFP-pulldown experiments, but only moderately decreased (35 ± 10%; means ± SEM, n = 6) its binding to PP1 (Fig. 4h). Conversely, alanine mutation of EGFP-SDS22 at the seven implicated basic residues (SDS22-KR7A) abolished its binding to I3 in EGFP-trapping assays and, as further discussed below, even considerably increased its interaction

with PP1 (Fig. 4i). While I3-DE4A and SDS22-KR7A showed a reduced interaction with SDS22-WT and I3-WT, respectively (Fig. 4h, i), their interaction was re-established by reciprocal charge-reversal mutations of both I3 (I3-DE4R) and SDS22 (SDS22-KR7E) (Fig. 4j). Finally, we performed ITC binding assays with purified proteins, which confirmed a direct interaction between SDS22 and His-I3 (K$_d$ = 7.6 ± 5.5 μM). However, their binding affinity was much reduced (K$_d$ = 127.6 ± 28.1 μM) by mutation of the I3-interaction site of SDS22 (Fig. 4k and Supplementary Fig. 4g). Together, these data firmly established a direct, ionic interaction site between SDS22 and I3.

The seven basic residues of SDS22 that interact with I3 have also been implicated in the binding of the splicing factor BCLAF1[18]. We confirmed that ectopically expressed SDS22-WT, but not SDS22-KR7A, binds to both I3 and BCLAF1 (Supplementary Fig. 4h). However, an ectopically expressed PP1–SDS22 fusion was associated with I3, but not with BCLAF1, in pulldown experiments. This indicates that there is no competition between I3 and BCLAF1 for binding to PP1–SDS22, probably because I3 is additionally anchored through PP1 binding. Hence, our data suggest that BCLAF1 only interacts with the cellular pool of SDS22 that is not associated with I3 and/or PP1.

## The SDS22:I3 interaction makes the recruitment of I3 to nascent PP1 irreversible

Next, we investigated the importance of the SDS22:I3 interaction for the recruitment of I3 to PP1. An SDS22-binding mutant of ectopically expressed mClover-tagged PP1γ (K147A, K150A)[18], referred to as PP1-2KA, was deficient in the binding of both SDS22 and I3 in mClover-trapping experiments (Fig. 5a), indicating that the efficient recruitment of I3 depends on SDS22. We also made use of a HeLa cell line that inducibly expressed Strep-tagged PP1γ[8] and found that I3 was no longer recruited to newly translated Strep-PP1γ (induction for 1 h) after the knockdown of SDS22 (Fig. 5b). Conversely, the recruitment of SDS22 to Strep-PP1γ was not affected by the knockdown of I3 (Fig. 5c). Finally, we used purified components to study how the binding of His-I3 to GST-PP1α is affected by SDS22 addition. The data confirmed that the PP1:I3 interaction was enhanced by SDS22 addition (Supplementary Fig. 5a) and that SDS22 addition rendered the interaction of I3 with GST-PP1, but not the SDS22-binding mutant GST-PP1-2KA, resistant to competitive disruption with NIPP1-(143–224) (Supplementary Fig. 5b). Together, these data demonstrated that SDS22 is recruited to nascent PP1 before I3 and that the efficient recruitment of I3, in the presence of RIPPOs that compete for binding to PP1, depends on its interaction with SDS22.

To further define the role of SDS22 in the recruitment of I3 to PP1, we made use of lysate-based split-luciferase assays[13]. These assays are based on the complementation between small (SmBiT) and large (LgBiT) fragments of Nanoluc luciferase (Promega). The SmBiT/LgBiT-fragments are catalytically inactive, but complement into active luciferase when brought in close proximity through interaction of their fused partners (Fig. 5d). To investigate how the I3:PP1 interaction is

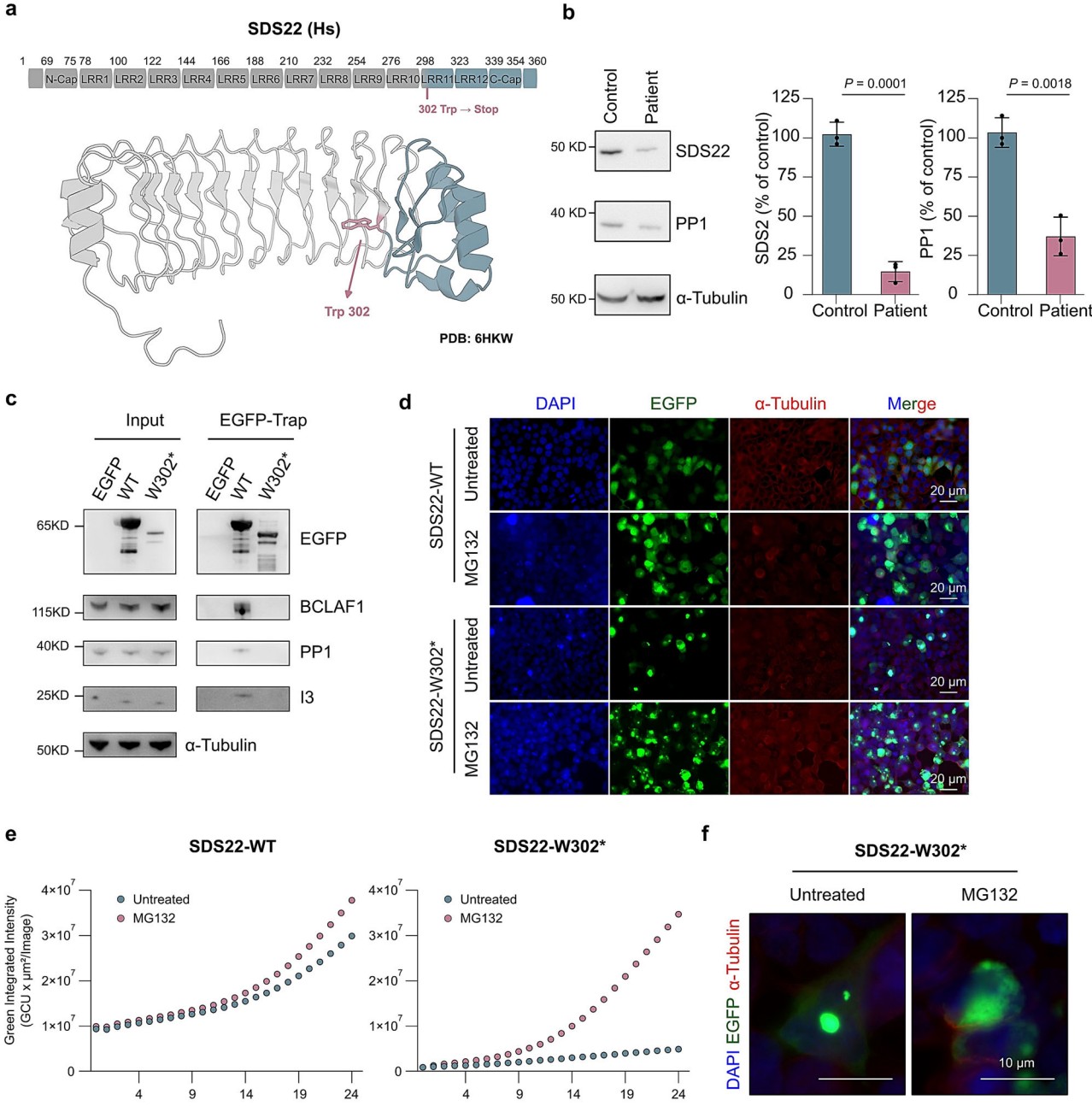

**Fig. 3 | Loss of SDS22 and PP1 in a human patient with neurodevelopmental disease. a** Domain structure of SDS22 (upper panel) and mapping of the heterozygous SDS22-W302* mutation in patient P1 (lower panel), predicted to result in the expression of an SDS22 variant that lacks LRR11, LRR12 and the C-terminal LRR-Cap (C-Cap). The blue-colored fragment is missing in SDS22-W302*. **b** Levels of PP1, SDS22 and α-tubulin in fibroblasts from patient P1 and four healthy controls, as detected by immunoblotting using antibodies that bind to the N-terminus of SDS22 or the C-terminus of all PP1 isoforms (left panel). The right panel shows the quantification of the immunoblotting data. The results are shown as means ± SD (n = 3 independent experiments). The control value in each experiment was the average for four controls and the values for one control were set at 100%. *P* values are from two-sided unpaired t-test. **c** Co-immunoprecipitation (EGFP-traps) of endogenous BCLAF1, PP1 and I3 with transiently expressed EGFP, EGFP-SDS22-WT (WT) or EGFP-SDS22-W302* (W302*) in HEK293T cells. **d** Immunofluorescence staining of HEK293T cells transiently transfected with expression vectors for EGFP-SDS22 or EGFP-SDS22-W302*. The cells were treated with or without 10 μM MG132 for 8 h before fixation. The fixed cells were stained for DNA (DAPI), EGFP and α-tubulin. The scale bars are 20 μm. **e** The effect of MG132 on the levels of EGFP-SDS22 and EGFP-SDS22-W302*, as measured in 1 h intervals by IncuCyte live-cell analysis for 24 h. HEK293T cells were transiently transfected with EGFP-SDS22-WT or EGFP-SDS22-W302*. 10 μM MG132 was added 1 h before the first measurement. Green fluorescence (Green Calibrated Unit, GCU) was measured to show EGFP intensity. **f** EGFP-SDS22-W302* accumulates in granules. Detail of cells shown in (**d**).

affected by SDS22, we generated lysates from HEK293T cells transiently transfected with constructs encoding either wildtype I3-SmBiT (I3^WT-SmBiT), I3-SmBiT with a mutated PP1-binding RVxF motif (I3^RATA-SmBiT), LgBiT-PP1α or LgBiT linked to a fusion of PP1α and SDS22 (LgBiT-PP1α-SDS22) (Fig. 5e, f). Luciferase complementation was obtained upon mixing of lysates with I3^WT-SmBiT and LgBiT-PP1α, but not with a mixture of lysates containing I3^RATA-SmBiT and LgBiT-PP1α (Fig. 5g), demonstrating that luciferase complementation depended on the binding of I3 to PP1. Interestingly, complementation with I3-SmBiT^WT was nearly threefold higher with lysates expressing LgBiT-

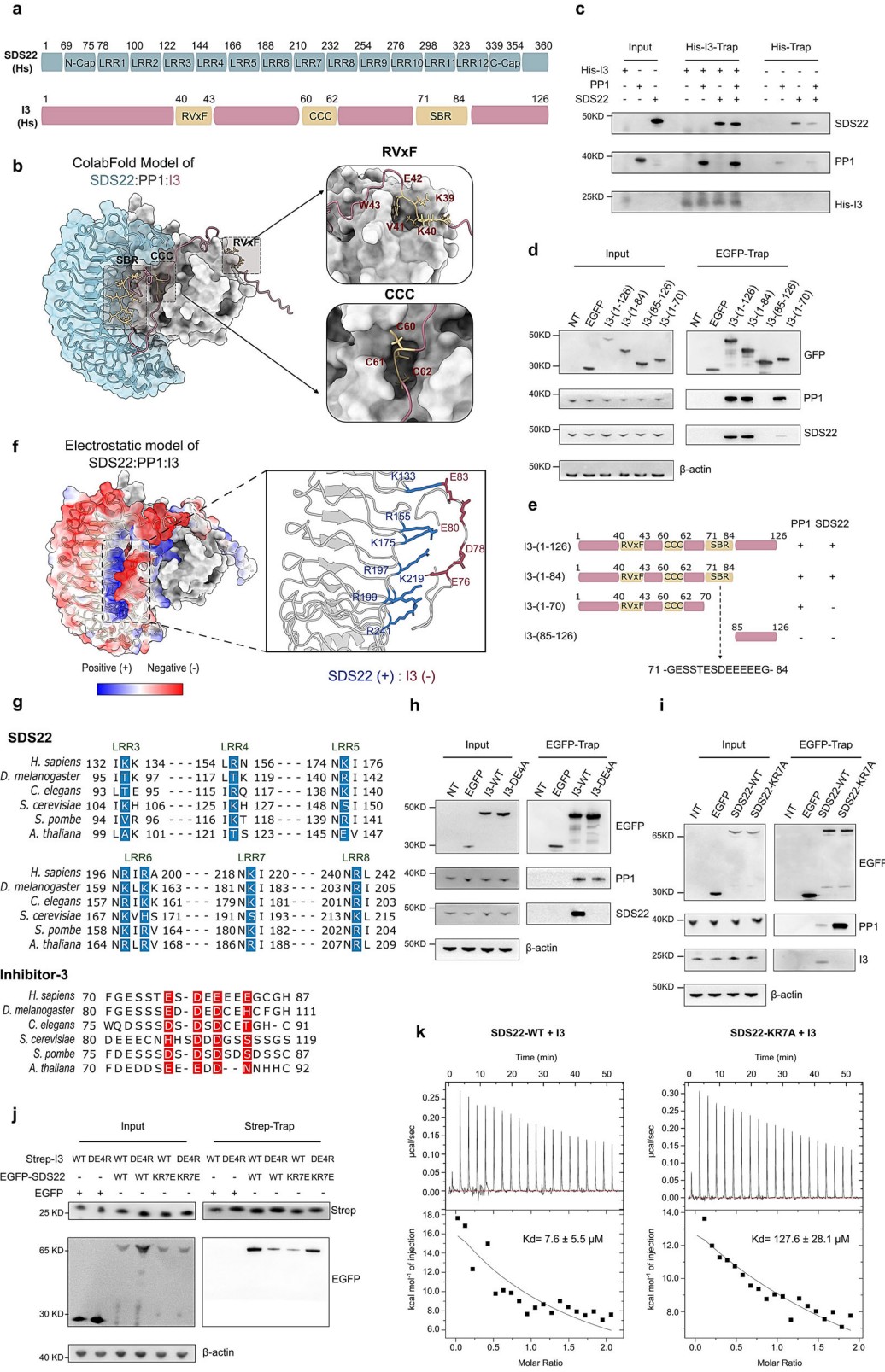

PP1α-SDS22 rather than LgBiT-PP1α (Fig. 5h), at similar expression levels (Fig. 5f). These data suggested that I3 binds with a higher affinity to a PP1–SDS22 fusion than to PP1, confirming a key contribution of the I3:SDS22 interaction. Importantly, the increased complementation with the SDS22-PP1 fusion was specific for I3, as it was much less pronounced with RepoMan (RM), a structurally unrelated RIPPO with a PP1-binding RVxF motif (Fig. 5e–h). The latter data are in accordance

with the ability of RM to form a ternary complex with SDS22:PP1 without, however, making direct contacts with SDS22[18]. The interaction between I2-SmBiT and LgBiT-PP1α was even reduced when PP1α was fused with SDS22 (Fig. 5e–h), consistent with the overlapping binding sites of SDS22 and I2 on PP1[9,18].

To delineate the effect of SDS22 on the dynamics of the PP1:I3 and PP1:RepoMan interactions, we performed kinetic experiments using

**Fig. 4 | Mapping of an SDS22:I3 interaction interface. a** Domain structure of SDS22 and I3. Shown are the LRR and C-Cap structures of SDS22, the PP1-binding RVxF- and CCC-motifs of I3 as well as the predicted SDS22-binding region (SBR) of I3. **b** The rank-1 AlphaFold-Multimer model (out of 25 predicted models) of SDS22:PP1:I3, using the amber-relaxation setup option. Residues with pLDDT scores <30 are not displayed. SDS22 (Hs), blue; PP1α (Hs), gray; I3 (Hs), pink. The PP1-binding motifs of I3 are highlighted with dashed boxes and are also shown as zoom-in views. **c** Immunoblotting of trapped His-I3 for the presence of PP1α and SDS22. His-I3 (2 µM) or His were incubated with buffer, PP1 (0.3 µM), SDS22 (0.3 µM) or PP1 + SDS22, and trapped by Ni-NTA magnetic beads. **d** Co-immunoprecipitation (EGFP-traps) of endogenous PP1 and SDS22 with ectopically expressed EGFP or EGFP-I3 variants, i.e., EGFP-tagged I3-(1–126; WT), I3-(1–84), I3-(1–70) or I3-(85–126). NT no transfection. **e** Schematic representation of the used EGFP-I3 truncation mutants and their interaction with PP1 and SDS22, as derived from data in (**d**). **f** Electrostatic surface of the SDS22:PP1:I3 AlphaFold model. The zoom-in view shows the charged residues of SDS22 and I3 that are predicted to interact. **g** Sequence alignment of the interacting fragments of SDS22 and I3, with the same color code for the involved charged residues. The alignment was generated using the EMBL-EBI MAFFT tool with the following UniProt identifiers: *Homo sapiens*, Q15435; *Drosophila melanogaster*, Q9VEK8; *Caenorhabditis elegans*, P45969; *Saccharomyces cerevisiae*, P36047; *Schizosaccharomyces pombe*, P22194; *Arabidopsis thaliana*, Q84WJ9. **h** Association of endogenous PP1 and SDS22 with transiently expressed, EGFP-trapped EGFP-I3-WT or EGFP-I3-DE4A. **i** Binding of endogenous PP1 and I3 to transiently expressed, EGFP-trapped EGFP-SDS2-WT or EGFP-SDS22-KR7A. **j** Co-immunoprecipitation (Strep-traps) of transiently expressed variants of Strep-I3 (Strep-I3-WT; Strep-I3-DE4R) and EGFP-SDS22 (EGFP-SDS22-WT; EGFP-SDS22-KR7E). The lower part of the EGFP blot (middle panel) was used for Strep-I3 detection (upper panel). **k** Isothermal titration calorimetry (ITC) experiments of purified SDS22-WT or SDS22-KR7A and His-I3. 5 µM SDS22-WT or SDS22-KR7A was titrated with 50 µM His-I3. The inset shows average $K_d$ values ± SEM (n = 3).

the split-luciferase system. Following the time-dependent assembly of PP1:I3 or PP1:RepoMan upon mixing of lysates containing the appropriately tagged subunits, both complexes were competitively disrupted by the addition of an excess of a PP1-binding RVxF-peptide (Fig. 5i and Supplementary Fig. 5c, d). However, when I3 and RepoMan complexes were made with a PP1–SDS22 fusion, only the RepoMan complex could still be disrupted by addition of an RVxF-peptide (Fig. 5j and Supplementary Fig. 5c), indicating that the binding of I3 had become irreversible by its simultaneous binding to PP1 and SDS22. In accordance with this interpretation, an SDS22-binding mutant of I3 (I3-DE4A) could be dissociated from the PP1–SDS22 fusion (Fig. 5k and Supplementary Fig. 5c) with an RVxF-peptide. Importantly, the interaction between LgBiT-PP1 and SDS22-SmBiT was barely affected by the addition of a large excess of untagged SDS22 (Fig. 5l), but untagged SDS22 did prevent the time-dependent formation of new complexes between the tagged fusions. In contrast, the addition of a PP1-binding mutant of SDS22 (SDS22-E192) did not prevent the assembly of new PP1:SDS22 complexes. These data demonstrated that SDS22 also binds irreversibly to PP1, consistent with affinity measurements[9]. Hence, both SDS22 and I3 bind irreversibly to PP1, explaining why their extraction from SDS22:PP1:I3 requires energy, as provided by p97/VCP-mediated ATP hydrolysis.

## The SDS22:I3 interaction is essential for PP1-holoenzyme assembly

Finally, we examined the importance of the direct interaction between SDS22 and I3 in the SDS22:PP1:I3 complex for the subsequent steps of PP1-holoenzyme assembly. The binding of EGFP-I3 to p97/VCP in EGFP-trapping experiments was lost after mutation of either its PP1-binding site (I3-KAEA) or SDS22-binding site (I3-DE4A) (Fig. 6a). This indicated that p97/VCP is recruited by SDS22:PP1:I3, but not by PP1:I3 or SDS22:I3, consistent with recent data showing that p97/VCP binds to both SDS22 and I3[12]. We also found that traps of an EGFP-PP1-SDS22 fusion contained I3, but relatively little of other SLiM-based RIPPOs, such as RepoMan or MYPT1 (Fig. 6b). This suggested that the irreversible recruitment of I3, resulting from its simultaneous binding to SDS22 and PP1, precludes the competitive binding of other RIPPOs. In accordance with this interpretation, an EGFP-PP1-SDS22 fusion with the SDS22 moiety mutated in its I3-binding site (EGFP-PP1-SDS22-KR7A) did co-precipitate canonical RIPPOs such as MYPT1 and Repo-Man. We furthermore found that SDS22-WT, ectopically expressed in HeLa cells, only interacted weakly and transiently with newly translated Strep-PP1γ (Fig. 6c), consistent with the gradual p97/VCP-mediated transfer of Strep-PP1γ to canonical RIPPOs[8]. In contrast, SDS22-KR7A interacted more strongly and for a prolonged time with nascent Strep-PP1 (Fig. 6c), accounting for its much increased complexation with the global pool of PP1 (Fig. 4i).

The above data indicated that the SDS22:I3 interaction is required for their co-extraction by p97/VCP during PP1-holoenzyme assembly.

Since SDS22-associated PP1 is inactive[9], the erroneous transfer of PP1:SDS22-KR7A to canonical RIPPOs is expected to generate non-functional holoenzymes. To test this hypothesis, we compared the phenotype of stable HeLa Flp-In cell lines that inducibly express either SDS22-WT or SDS22-KR7A (Fig. 6d). While the overexpression of SDS22-WT had no effect on proliferation, the overexpression of SDS22-KR7A resulted in a reduced proliferation (Fig. 6e). This proliferation-deficit was not caused by a slower progression from $G_1/S$ to $G_2/M$, as shown by flow-cytometry analysis of cells that were released from a $G_1/S$ arrest (Supplementary Fig. 6a). However, cells expressing SDS22-KR7A needed more time for the completion of mitosis, as illustrated by flow-cytometry analysis of cells released from a $G_2/M$ arrest with RO3306 (Fig. 6f and Supplementary Fig. 6b) as well as by time-lapse video imaging (Fig. 6g). The mitotic-arrest phenotype of SDS22-KR7A expression correlated with an increased phosphorylation of established mitotic PP1 substrates, including ERM proteins and histone H3 at Thr3 (Fig. 6h). Together, these data demonstrate that a failure to co-extract SDS22 and I3 from SDS22:PP1:I3 results in the formation of PP1 holoenzymes that contain SDS22 as a third, inhibitory subunit.

## Discussion

We have generated cellular and molecular research tools to address remaining key questions on the biogenesis of PP1 holoenzymes (see "Introduction"), with a particular focus on the role of SDS22. The available data allow us to propose a more detailed model on how nascent PP1 is transferred, in a stepwise manner, to canonical RIPPOs, thereby forming functional holoenzymes (Fig. 7). Newly translated PP1 first recruits SDS22 (Fig. 5b, c). The irreversible nature of SDS22 recruitment probably stems from the large number of contacts between PP1 and the highly structured PP1-binding domain of SDS22[9,18]. Moreover, SDS22 locks PP1 in inactive conformation that precludes metal-1 loading. SDS22 recruitment serves to stabilize nascent PP1 (Fig. 2h, i), which is prone to aggregation in its unbound state[9,25]. In the absence of SDS22, newly translated PP1 is gradually lost (Figs. 2h, i and 7), resulting in the hyperphosphorylation of PP1 substrates and a nearly complete proliferation arrest (Figs. 1c and 2c, e, f). The next step in the biogenesis of PP1 involves the recruitment of I3 to SDS22:PP1 (Figs. 5b, c and 7), which prevents substrate binding due to occlusion of the active site by the CCC motif of I3[6]. The binding of I3 is also irreversible because of its simultaneous interaction with SDS22 and PP1 (Fig. 5i–k). This irreversibility explains the selectivity of I3 recruitment, as it precludes competition with other SLiM-based RIP-POs for binding to SDS22:PP1 (Fig. 6b), as well as the energy-dependency of its extraction during holoenzyme assembly. A first glance, our data on the stabilization of PP1:I3 by SDS22 (Fig. 5i, j and Supplementary Fig. 5c) are at variance with a recent study indicating that SDS22 weakens the PP1:I3 interaction[26]. However, the latter study was performed with N-terminally truncated PP1 and SDS22, which may affect their interaction with I3.

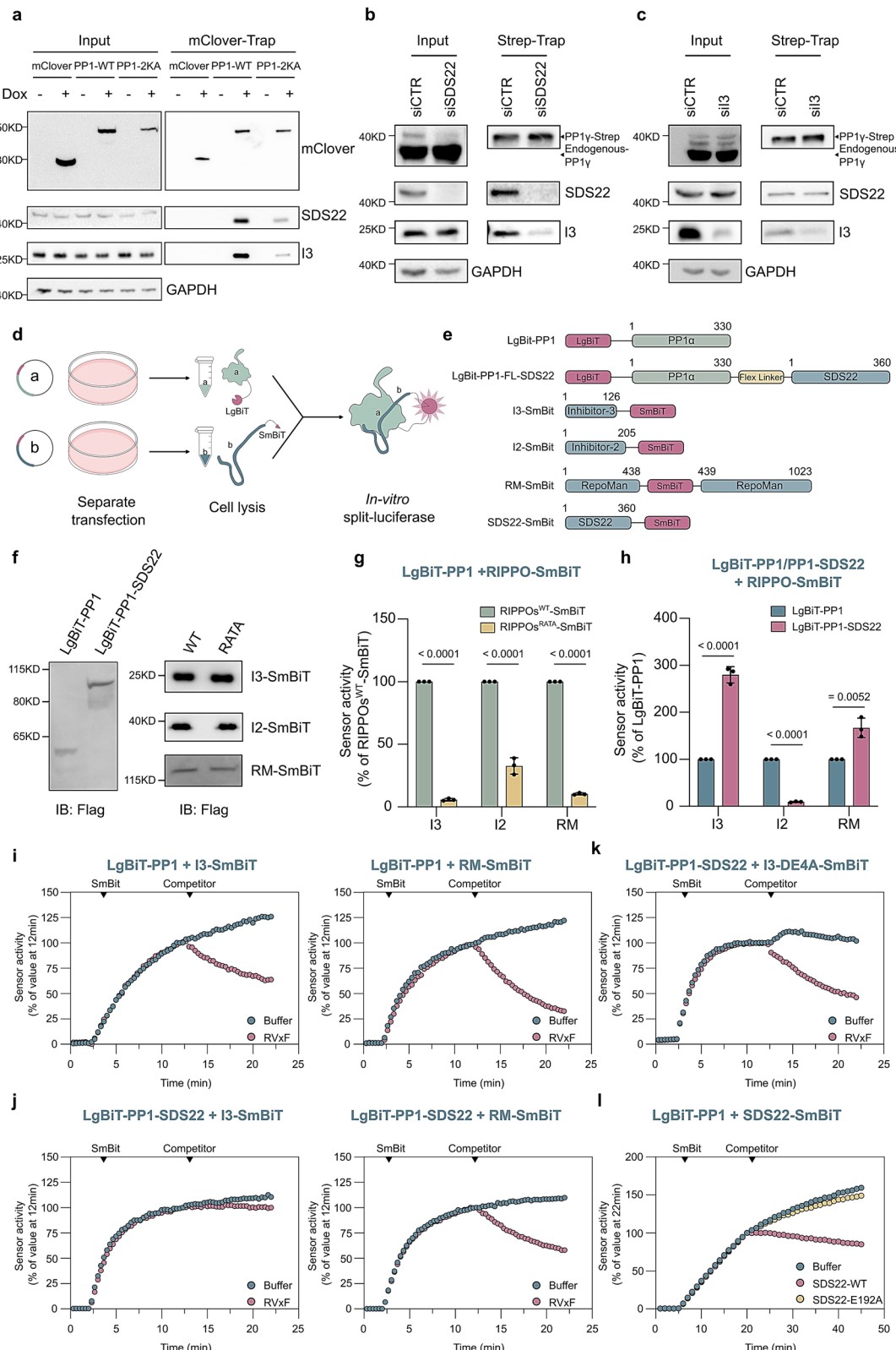

It is only when the ternary SDS22:PP1:I3 complex is formed that p97/VCP is efficiently recruited (Figs. 6a and 7), consistent with the recent report that p97/VCP interacts with both SDS22 and I3[12]. The p97/VCP-catalyzed unfolding of I3 results in the co-extraction of SDS22[8], which is critically dependent on its interaction with I3 (Fig. 6b). SDS22 that is mutated in its I3-interaction site erroneously co-transfers with PP1 to canonical RIPPOs, resulting in the formation of inactive PP1

holoenzymes that contain SDS22 as an inhibitory third subunit (Figs. 6b, h and 7). SDS22 may be co-extracted with I3 because of the pulling forces exerted by I3 unfolding in the p97/VCP-hexamer channel[8]. In addition, the co-extraction of SDS22 may be facilitated by the (hypothetical) coupled incorporation of metal 1 (Zn$^{2+}$) in the active site, which is known to reduce the binding affinity of SDS22 for PP1[9]. The incorporation of Zn$^{2+}$ possibly involves I3, as suggested by its

**Fig. 5 | The recruitment of I3 to nascent PP1 depends on SDS22. a** Association of mClover-trapped mClover, mClover-PP1-WT or mClover-PP1-2KA with endogenous SDS22 and I3. The expression of mClover-PP1 fusions in HeLa FlpIn T-REx cell lines was induced for 24 h with Dox. **b** Reduced recruitment of I3 to nascent Strep-PP1 (Strep-trapping) following the knockdown of SDS22. Strep-PP1γ expression in a HeLa FlpIn T-REx cell line was induced for 1 h with Dox. **c** The recruitment of SDS22 to nascent Strep-PP1 (Strep-trapping) was not affected by the knockdown of I3. The cells were treated as detailed for (**b**). **d** Overview of the lysated-based split-luciferase protocol. The interacting proteins (a and b) are shown in blue and the fused luciferase fragments are shown in pink. **e** Scheme of the used fusions with LgBiT or SmBiT. **f** Transient expression of SmBiT/LgBiT fusions in HEK293T cells. **g** Luciferase complementation following mixing of HEK293T lysates containing LgBiT-PP1 and a SmBiT-tagged RIPPO (WT or RATA mutant). The signals were normalized for the same expression level. The results were plotted as a percentage of the signal with LgBiT-PP1α + SmBIT-RIPPO[WT] (means ± SD; n = 3 independent experiments). *P* values were from two-sided unpaired t-test. **h** Comparison of luciferase complementation with the indicated SmBiT-RIPPOs (WT) and either LgBiT-PP1 or LgBiT-PP1–SDS22. The results were plotted as a percentage of the signal with LgBiT-PP1 + SmBIT-RIPPO. **i** Kinetic-trace experiments showing the time-dependent association of LgBiT-PP1 with I3-SmBiT or RM-SmBiT after addition of the RIPPO-SmBiT fusions (SmBiT). Also shown is the competitive disruption of the assembled complexes with 5 μM NIPP1-(143–224). The data are plotted as a percentage of the signal just before addition of competitor at 12 min. **j** Same experiment as in (**i**) but with LgBiT-PP1-SDS22. **k** Same experiment as in (**j**) but with I3-DE4A-SmBiT. **l** Kinetic-trace experiment showing the time-dependent association of LgBiT-PP1α and SDS22-SmBiT. Purified SDS22-WT and SDS22-E192A (10 μM) were added as competitors. The data are plotted as a percentage of the signal just before addition of competitor (t = 22 min). The experiments shown in (**i–l**) are representative for three independent experiments.

recent identification as a Zn$^{2+}$-binding protein[6]. However, the mechanistic details of metal loading in the active site of PP1 are still unresolved. A final step in the biogenesis of PP1 holoenzymes, associated with the co-extraction of I3 and SDS22 from SDS22:PP1:I3, is the (spontaneous) association of liberated PP1 with canonical RIPPOs (Fig. 7). As RIPPOs are expressed in a large molar excess over PP1[2], all p97/VCP-freed PP1 will be rapidly titrated, which precludes uncontrolled dephosphorylation by the free catalytic subunit[27]. The diversity of PP1:RIPPO complexes that accumulate in the cell is largely determined by the relative concentration of RIPPOs and their affinity for PP1, which are both tightly regulated. For example, the affinity of many RIPPOs for PP1 is reduced during the first half of mitosis by phosphorylation of residues within or close to the RVxF motif[28]. As SLiM-based PP1:RIPPO interactions are highly dynamic, changes in the expression of RIPPOs or in their affinity for PP1 result in a spontaneous re-equilibration between PP1 holoenzymes.

The proposed model of the biogenesis of PP1 holoenzymes (Fig. 7) reconciles a large body of data in the literature. Thus, the model explains why SDS22 is an inhibitor of PP1 in biochemical assays but an activator of PP1 in a cellular context (for references see ref. 5), as SDS22 inhibits nascent (or purified) PP1 but also coordinates holoenzyme assembly. The phenotypic analysis of cells lacking or overexpressing SDS22 has led to the conclusion that SDS22 is implicated in the targeting of PP1 to subcellular complexes such as kinetochores[29], Aurora B[30] and ERM proteins[31,32]. However, our data strongly suggest that the role of SDS22 is indirect and that a lack of SDS22 leads to a loss of PP1 and a general depletion of PP1 holoenzymes. Conversely, a disturbance in the SDS22:I3 balance by either SDS22 overexpression[18,32] or I3 depletion[33] results in the inhibitory association of SDS22 with canonical PP1:RIPPO complexes. However, the phenotype of SDS22-WT overexpression is less severe than that of SDS22-KR7A expression because SDS22-WT can still be (partially) co-extracted with I3 (Fig. 6). Finally, the model explains why depletion of SDS22 or PP1 has a similar phenotype[34], as SDS22 depletion culminates in a loss of (newly translated) PP1. In contrast, I3 is not needed for the stabilization of newly translated PP1. In the absence of I3, newly translated PP1 may even be used for holoenzyme assembly in a p97/VCP-independent manner, as its RVxF-docking site is accessible for the binding of canonical RIPPOs. This may explain why a short-term, siRNA-mediated depletion of I3 has a less severe proliferation phenotype than SDS22 depletion (Supplementary Fig. 1e). However, a permanent depletion of I3 is not viable for human cells (https://depmap.org/portal/), which likely stems from a gradual loss of (functional) PP1 holoenzymes.

The deletion of SDS22 has been associated with various cell-division defects, including a metaphase–anaphase arrest (Figs. 1, 2 and ref. 17), reduced Aurora-B inactivation[29,30], and incomplete ERM dephosphorylation (Fig. 2 and ref. 32). Our data suggest that these phenotypes are all caused by the associated loss of PP1, resulting in a deficiency of PP1 holoenzymes that are essential for progression

through mitosis. A similar mitotic phenotype is observed following the competitive disruption of PP1 holoenzymes by a massive overexpression of a canonical RIPPO like NIPP1[35]. While these data all validate the essential role of SDS22 and PP1 in cell division, they do not exclude key functions in other cellular processes such as replication, which, however, are masked by the cell-cycle arrest that is induced by the depletion of SDS22 or PP1.

Unexpectedly, we found that a patient suffering from a severe neurodevelopmental disease bears a mutated *PPP1R7* allele in heterozygosis, resulting in the expression of an unstable SDS22 variant (Fig. 3b–f). The loss of SDS22 in this patient correlated with a decreased expression level of PP1 (Fig. 3b), as also detected in the SDS22-degron cell line (Fig. 2h, i). We have not yet been able to identify additional patients with mutant *PPP1R7* alleles, nor to obtain an animal model, which will be necessary to establish a firm causal relationship between the loss of SDS22 and PP1, and the development of neurodevelopmental disease. However, it is worth noting that loss-of-function mutations of PP2A, a close structural relative of PP1, have already been established as a cause of neurodevelopmental disorders[36]. Further exploration of the role of SDS22 and PP1 in the development of neurodevelopmental disorders is important, as it may provide avenues to treat disease progression, for example using inhibitors of protein kinases that act antagonistically to PP1, thereby restoring the phosphorylation balance of PP1 substrates.

In conclusion, our data identified SDS22, 33 years after its initial discovery[17], as a key integrator of the major steps of PP1-holoenzyme assembly (Fig. 7). The essential role of SDS22 in the biogenesis of PP1 is entirely consistent with its early evolutionary origin, shortly after the emergence of PP1 and before the divergence of the supertaxa that constitute the eukaryotic crown[22].

## Methods

### Ethics
This study was performed according to the guidelines of the Ethics Committee Research (EC research) UZ/KU Leuven (S63808). The study design and conduct complied with all relevant regulations regarding the use of human study participants and was conducted in accordance to the criteria set by the Declaration of Helsinki.

### Materials
The sequence of all new constructs generated for this study is indicated in Supplementary Data 4. The source of all key materials (chemicals, constructs, antibodies, siRNAs, cell lines) and adopted software is indicated in Supplementary Data 5.

### Protein expression and purification
His-I3 and His-NIPP1-(143–224), cloned into the pET16b expression vector, were expressed in bacterial BL21 cells at 37 °C in Luria Bertani broth, fortified with 100 μg/ml ampicillin, until an optical density at

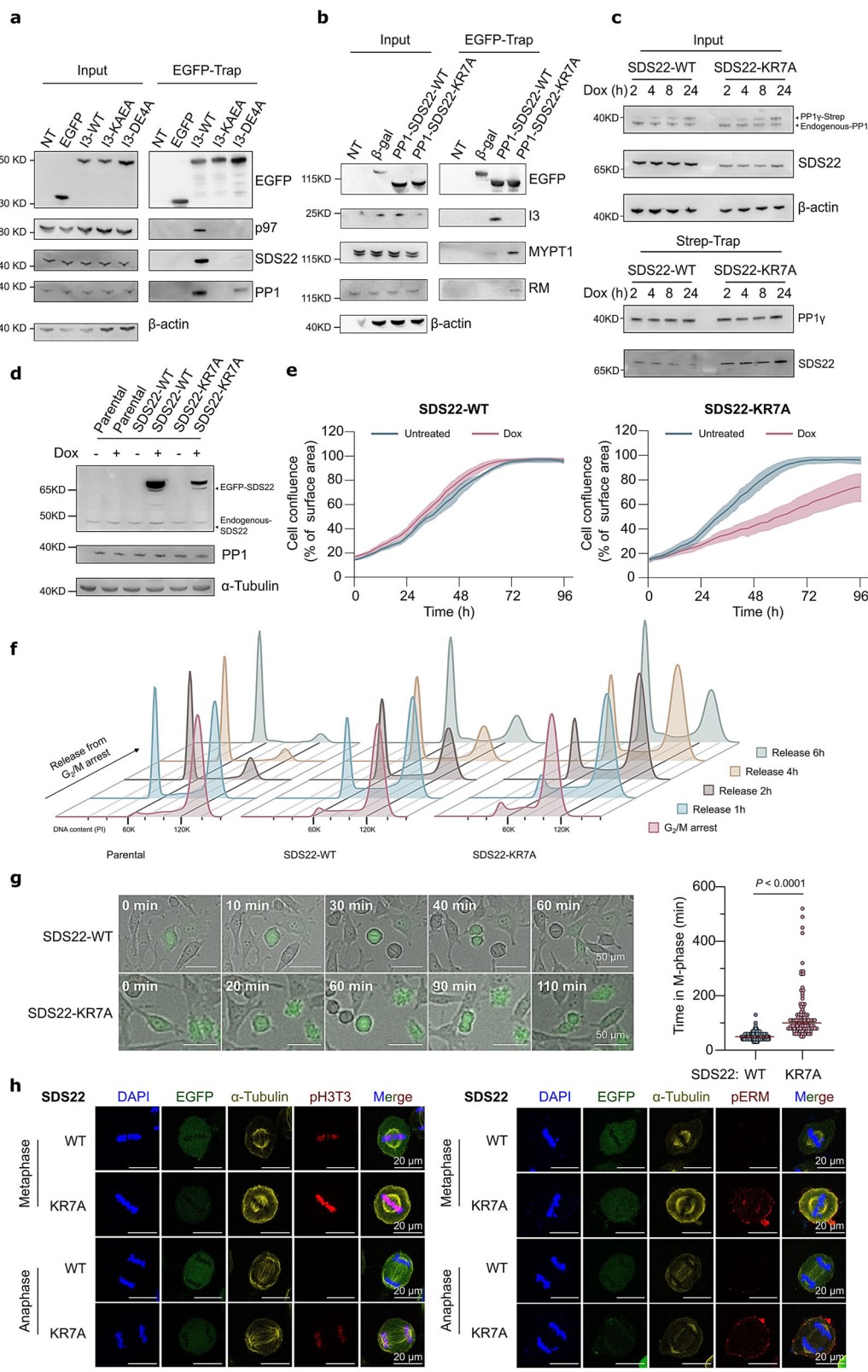

600 nm of 0.6. Protein expression was induced with 1 mM isopropyl β-D-1-thiogalactopyranoside (IPTG) for 20 h at 18 °C. The cells were harvested by centrifugation (6000 × g for 15 min), resuspended in lysis buffer, comprising 20 mM Tris at pH 7.9, 0.5 M NaCl, 0.5% Triton X-100, 0.5 mM benzamidine and 0.5 mM leupeptin, and lysed by a single freeze-thaw cycle, followed by 15 min sonication with a Diagenode Bioruptor. The lysates were subjected to centrifugation (20 min at

15,000 × g), and the supernatant was loaded onto Ni²⁺-Sepharose beads for 2 h at 4 °C. The beads were washed with a buffer containing 20 mM Tris at pH 7.9, 50 mM imidazole and 0.5 M NaCl. Finally, the proteins were eluted with 20 mM Tris at pH 7.9, 0.4 M imidazole and 0.5 M NaCl, and stored at −80 °C. GST-PP1α and tag-free PP1α (generated from intein-tagged PP1α[37]) were purified with the same protocol as described for His-tagged proteins, but using glutathione agarose/

**Fig. 6 | The SDS22:I3 interaction is essential for the assembly of functional PP1 holoenzymes. a** Association of endogenous p97/VCP, SDS22 and PP1 with transiently expressed and trapped EGFP or EGFP-trapped I3 variants from HEK293T cell lysates. NT non-transfected. **b** Co-immunoprecipitation of endogenous I3, MYPT1 and RepoMan (RM) with transiently expressed and trapped EGFP-tagged PP1-SDS22-WT/KR7A or EGFP-β-galactosidase (negative control) from HEK293T cell lysates. **c** Co-immunoprecipitation of transiently expressed EGFP-tagged SDS22-WT or SDS22-KR7A with Strep-PP1γ, induced in HeLa FlpIn T-REx cells with Dox for 0→24 h. **d** Expression of EGFP-tagged SDS22-WT and SDS22-KR7A in HeLa FlpIn T-Rex cell lines after Dox addition for 48 h. Also shown is endogenous SDS22 and PP1. **e** Cell proliferation of HeLa FlpIn T-Rex cells, treated or not with Dox to induce the expression of EGFP-tagged SDS22-WT or SDS22-KR7A. Cell proliferation was measured in 3 h intervals using IncuCyte. Cell confluency is shown as a percentage of the surface. The solid line represents the means of three technical replicates and the shaded area represents the SD range from a single data set, which is representative for three independent experiments. **f** The effect of EGFP-tagged SDS22-WT or SDS22-KR7A expression (induction for 48 h) on cell-cycle progression. The cells were first arrested at the $G_1$/S transition. After thymidine release, the cells were blocked at the $G_2$/M transition with RO3306. After RO3306 washout for the indicated times, the cells were fixed, stained with propidium iodide and analyzed by flow cytometry. **g** Images (left panel) and quantification (right panel) of M-phase duration, as derived from live-cell imaging of cells induced (48 h) to express EGFP-tagged SDS22-WT or SDS22-KR7A. The cells were analyzed after a double-thymidine arrest and a release for 6 h. *P* value is from two-sided unpaired t-test (n = 100 cells). The scale bars are 50 μm. **h** Confocal images of cells induced to express EGFP-tagged SDS22-WT or SDS22-KR7A for 48 h. Cells were treated with SDS22 siRNA. The cells were fixed following a release (60 min) from an RO3306 block. Cells in metaphase and anaphase were stained for DAPI, EGFP, α-tubulin, histone H3 phosphorylated at Thr3 (pH3T3) and pERM. Scale bars, 20 μm.

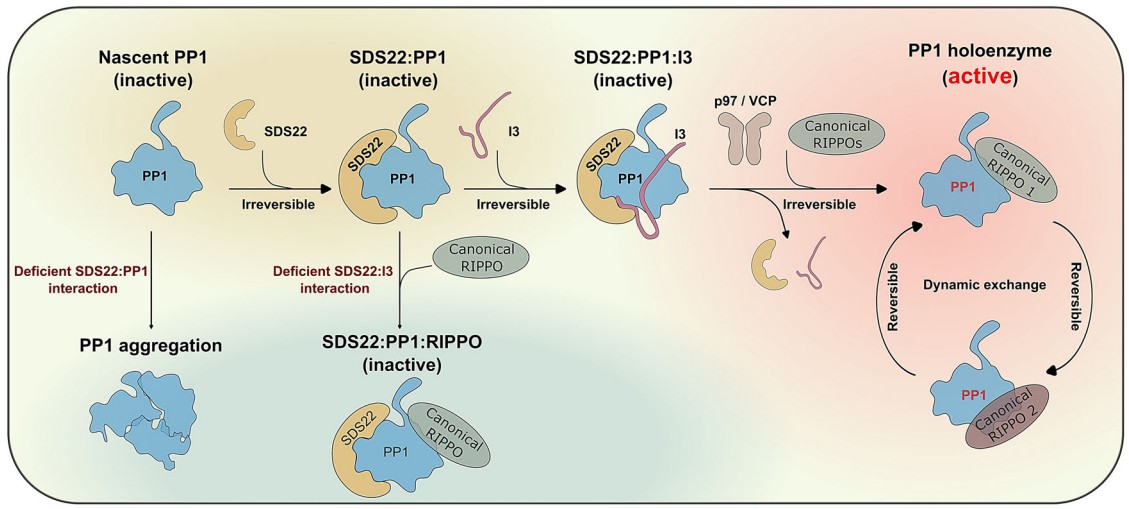

**Fig. 7 | Model of the stepwise biogenesis of PP1 holoenzymes.** SDS22 binds irreversibly to newly translated PP1 to prevent its aggregation. SDS22:PP1 recruits I3 in an irreversible manner, due to the simultaneous interaction of I3 with PP1 and SDS22. Subsequently, SDS22 and I3 are co-extracted from the ternary SDS22:PP1:I3 complex by p97/VCP, enabling the association of released PP1 with canonical RIPPOs to form functional holoenzymes. In the absence of I3 or after mutation of the SDS22:I3 interaction site, SDS22 and associated PP1 are co-transferred to canonical RIPPOs, resulting in the assembly of inactive SDS:PP1:RIPPO complexes.

chitin beads for affinity purification. The beads were washed with 20 mM Tris at pH 7.9, 0.5 M NaCl and 0.5 mM leupeptin. The proteins were eluted with 20 mM Tris at pH 7.9, 0.5 M NaCl, supplemented with 10 mM glutathione (for GST-PP1α elution) or 100 mM DTT (for cleavage of the intein tag and elution of tag-free PP1α), and stored at −80°C. pCHWMS-EGFP-IRES-puro TEV site-SDS22 was previously described[18] and transiently transfected in HEK293T cells. EGFP-tagged SDS22 was affinity-purified from the supernatant using EGFP-trap beads. Tag-free SDS22 was obtained by an overnight incubation with a TEV protease.

## Cell culture and treatments
HEK293T cells were cultured in Dulbecco's modified Eagle's medium with 4.5 mg glucose/ml, and supplemented with 10% fetal calf serum, 100 units penicillin/ml, and 100 μg streptomycin/ml. Fibroblast cell lines were cultured in the same medium, but with only 1 mg glucose/ml. HeLa Flp-In T-REx cell lines were also cultured with low-glucose medium and 100 units penicillin/ml, 100 μg streptomycin/ml, 200 μg hygromycin/ml and 5 μg blasticidin/ml. HeLa Flp-In T-REx cells were treated with Dox (1 μg/ml) to induce the expression of EGFP-SDS22 variants or Strep-PP1 variants. Transient overexpressions were performed using JetPRIME transfection reagent for 48 h. Transient knockdowns of SDS22, I3 or Mad2 (see Supplementary Data 5 for details) were carried out with Lipofectamine RNAi$^{MAX}$ Transfection Reagent. To induce a $G_2$/M arrest, cells were consecutively treated with 2 mM thymidine (20 h), released from thymidine (4 h) and treated with 9 μM RO3306 (15 h). A double-thymidine arrest involved a treatment with 2 mM thymidine (2 × 15 h), with an intervening period of 9 h without thymidine.

## Generation of SDS22-degron cell line
All used primers and PCR conditions are detailed in Supplementary Data 5. Q5 High-Fidelity Polymerase was used to amplify the pMK290 backbone and mAID-Clover-Neo/Hyg fragments. The SDS22 5' and 3' homology arms were amplified using HCT116 Tet-OsTIR1 genomic DNA as a template. Equimolar ratios of the four fragments were ligated using NEBuilder. Following ligation, the plasmid was digested with BbvCI and a 4(G4S) spacer was inserted. The resulting plasmid was verified by sequencing and used as a template for the generation of the neomycin- and hygromycin-resistance donor DNA constructs. A chimeric guide RNA (gRNA)/Cas9 expressing plasmid was made by digestion of pX330 with BbsI and ligation with a phosphorylated duplex gRNA-generating sequence that targets Cas9 upstream of the endogenous SDS22 stop codon. HCT116 Tet-OsTIR1 cells were transfected with the hygromycin- and neomycin-donor DNA constructs and the gRNA-containing pX330 plasmid at a ratio of 2:2:1, using Lipofectamine 2000. Transformants were selected by culturing in hygromycin (200 μg/ml) and G418 (850 μg/ml), and individual colonies were isolated and expanded. HCT116 Tet-OsTIR1 cells express the TIR1 protein from rice (*Oryza sativa*) from both alleles of the AAVS1 locus

under the control of a tetracycline-inducible promoter and was used as parental cell line[14]. The dox/IAA-induced degradation of the SDS22-mAID-clover fusion protein was determined after the addition of Dox (2 μg/ml) and IAA (500 μM) for the indicated times.

### Flp-In T-REx HeLa stable cell line generation and treatment

HeLa Flp-In T-REx cells, generously provided by Stephen Taylor from Manchester University, were used to generate stable cell lines for the expression of EGFP-tagged SDS22-WT or SDS22-KR7A in a Dox-inducible manner, using the pCDNA5/FRT/TO vector. More specifically, the pCDNA5/FRT/TO vector with inserted EGFP-SDS22-WT or EGFP-SDS22-KR7A was co-transfected with pOG44 (encoding Flp recombinase) into the HeLa Flp-In host cell line. Following transfection, the cells were cultured in DMEM, supplemented with 10% tetracycline-free FBS, hygromycin (200 μg/ml), and blasticidin (5 μg/ml). Transgene expression was induced by adding Dox (100 ng/ml) to the culture medium. HeLa Kyoto and HeLa Flp-In PP1γ−2xStrep cell lines were received from Daniel W. Gerlich (Vienna Biocenter) and Hemmo Meyer (University of Duisburg-Essen), respectively.

### Immunoblotting and immunoprecipitation

Cells were collected by centrifugation for 5 min at $200 \times g$, and resuspended in lysis buffer containing 20 mM Tris at pH 7.9, 0.3 M NaCl, 0.5% Triton X-100, 100 mM PMSF, 5 μg/ml leupeptin, 50 mM NaF, 10 mM β-glycerophosphate and 100 mM vanadate. After sonication and centrifugation for 10 min at $18,000 \times g$, the protein concentration of the supernatant (cell lysate) was measured by Bradford assay. Input samples were prepared by diluting 5% of the supernatant to 1 μg/ml protein concentration in sample buffer. The remaining supernatant was incubated for 2 h at 4 °C with 30 μl of home-made anti-EGFP nanobodies coupled to agarose. After washing, the beads were boiled with sample buffer and analyzed by SDS–PAGE on NuPAGE® 4–12% Bis-Tris gels, followed by blotting on PVDF membrane. The immunoblots blots were visualized using ImageQuant LAS 4000. For the detection of PP1 levels after SDS22 depletion, the cells were lysed for 15 min with modified RIPA buffer comprising 20 mM Tris at pH 7.9, 150 mM NaCl, 1 mM EDTA, 1 mM EGTA, 1% NP40, 1% sodium deoxycholate, 1 mM PMSF, 5 μg/ml leupeptin, 25 mM NaF, 10 mM β-glycerophosphate and 1 mM vanadate. The lysed cells were subjected to centrifugation for 10 min at $18,000 \times g$, and the supernatant was used as "cell lysate".

### Split-luciferase assay

Lysate-based split-luciferase assays were performed as described by Claes & Bollen[13]. Briefly, luciferase activity was measured at room temperature, using a Luminoskan Ascent (Thermo Scientific), either at 20-s intervals for kinetic measurements or after 15 min for endpoint measurements. All used split-luciferase constructs are described in ref. 13 or detailed in Supplementary Data 5.

### Cell-confluency assays

For IncuCyte live-cell assays, SDS22-degron cells or HeLa Flp-In T-REx cells were plated in a 96-well plate at $7-8 \times 10^3$ cells/well. Confluency of the cultures was measured every 3 h using IncuCyte system (Essen Biosciences, Ann Arbor, MI) over 96 h in medium. For SRB assays, the cells were fixed with a 10% trichloroacetic acid and stained with SRB dye for 30 min. Excess dye was removed by washing the cells with 1% acetic acid. The protein-bound dye was solubilized in a 10 mM Tris-base solution. The optical density of the solubilized dye was measured at 450 nm using a microplate reader.

### Flow cytometry

For cell-cycle analysis with flow cytometry, cells were harvested by trypsinization, washed with PBS, and fixed overnight with 70% ethanol at −20 °C. Subsequently, the cells were washed twice with PBS, resuspended in PBS containing 0.05% Triton and 0.1 mg/ml RNAse, and

incubated for 30 min at 37 °C. Propidium iodide (PI) was added at a final concentration of 70 μg/ml. PI was added to the cell suspension 1 h prior to FACS. For the assay conducted in the HCT116 SDS22-degron cell line, singlet events were collected, the $G_2/M$ phase was gated and compared across various conditions. In the assay using the HeLa SDS22 Flp-In cell line, singlet events were also collected, and GFP-positive cells were gated. The $G_2/M$ phase was quantified and compared across each experimental group. Cell-cycle analysis was performed using a BD Canto II flow cytometer and quantification of the data was done using FlowJo software.

### Immunofluorescence

Cells were cultured in a 24-well plate on poly-lysine coated coverslips, then fixed with 4% paraformaldehyde and permeabilized with 0.5% Triton X-100. After blocking with 3% BSA in PBS, the coverslips were incubated overnight with primary antibody and for 60 min with secondary antibody. DAPI staining was used to label DNA and coverslips were mounted using Mowiol. High-resolution confocal images were captured using a Leica TCS SPE laser-scanning confocal system on a Leica DMI 4000B microscope, equipped with a Leica ACS APO 40 × 1.15NA oil objective. Quantification was performed using ImageJ.

### Time-lapse imaging and analysis

Cells were cultured in a 24-well plate with DMEM lacking phenol red to reduce autofluorescence. To facilitate time-lapse imaging, the Leica TCS SPE laser-scanning confocal microscope was used, furnished with a live-imaging chamber that upheld conditions of 37 °C and 5% $CO_2$. Additionally, a monochrome digital camera, specifically the DFC365 FX model from Leica, was integrated into the system. Utilizing epifluorescence and differential interference contrast (DIC) microscopy with a 20× objective, cells were subjected to imaging every 10 min. Subsequent to image acquisition, the sequences were exported for subsequent analysis within the ImageJ software platform.

### RNA sequencing and phosphoproteomics

Parental and SDS22-degron HCT116 cells were untreated or treated with Dox (2 μg/ml) plus IAA (500 μM) for 48 h prior to harvesting. RNA from triplicate samples was harvested using the GenElute™ Mammalian Total RNA Miniprep Kit from Sigma. Sequencing and data production and analysis was performed by the VIB Nucleomics Core (www.nucleomics.be). For phosphoproteomics, the cells were lysed with 50 mM Tris at pH 7.9, 8.5 M urea, 150 mM NaCl, 5 mM sodium molybdate, Roche EDTA-free protease inhibitor cocktail and Roche PhosStop, and stored at −80 °C. Protein concentration was determined by BCA assay (Pierce/Thermo Fisher Scientific). Proteins were then reduced with 5 mM DTT for 30 min at 5 °C, cooled to room temperature, and alkylated with 15 mM iodoacetamide for 45 min in the dark at room temperature. Alkylation reactions were quenched with an additional 5 mM of DTT. The lysates were diluted sixfold with 25 mM Tris at pH 8.1 before tryptic digestion with 1:100 (w/w) trypsin at 37 °C overnight. Next day, digests were quenched by acidification with TFA, centrifuged to remove precipitation, desalted and peptides were lyophilized. Phosphopeptide enrichment was carried out with the High SelectTM Fe-NTA Phosphopeptide Enrichment Kit (Thermo Fischer Scientific), following the manufacturer's instruction, and desalted. Phosphopeptides were resuspended in 100 mM HEPES at pH 8.5/20% ACN and the respective TMTpro reagent, followed by incubation for 1 h at room temperature. A sample was removed to determine labeling efficiency, while the remainder was stored at −80 °C. Labeling was confirmed to be at least 95% efficient before quenching, mixing, and offline pentafluorophenyl (PFA)-based reverse-phase HPLC fractionation[38]. TMTpro-labeled peptides were analyzed on an Orbitrap Lumos mass spectrometer (Thermo Scientific) equipped with an Easy-nLC 1200 (Thermo Scientific), and raw data was searched and processed as described by Smolen et al.[39]. Peptide intensities were

adjusted based on total TMT reporter ion intensity in each channel, and log2 transformed. Statistical analysis was performed in Perseus[40].

## Isothermal titration calorimetry (ITC)

Isothermal titration calorimetry (ITC) experiments were performed with a MicroCal iTC200 instrument (Fig. 4k). Each experiment was executed with newly purified proteins. The protein of lower concentration was placed in the cell (300 μl), while the protein of higher concentration (His-I3) was loaded by a syringe in 20 consecutive injections of 2 μl, except for the first injection (0.5 μl). The reaction temperature was maintained at 22 °C. The heat of dilution for the injectant, under the same conditions, was subtracted from the binding experiment before analysis of the binding. The concentration of injectant that could be used (50 μM His-I3) was limited by aggregation at higher concentrations, accounting for the non-saturation of the binding curves. The data were analyzed with the software provided with the MicroCal iTC200 instrument.

## AlphaFold-Multimer

DeepMind's AlphaFold-Multimer[23] was used to model the structure of SDS22:PP1:I3. Protein sequences were obtained from Uniprot and employed to generate multiple sequence alignments (MSAs). The models were generated using the Colab interface, with associated confidence scores (pLDDT) and visualized using PyMOL and UCSF ChimeraX.

## Analysis of patient P1

Patient P1 was enrolled in the URDCat (Undiagnosed Rare Diseases in Catalonia) project. Her parents provided written informed consent for genetic testing, release of clinical data and publication of anonymous photos. The clinical data set of P1 is described in the supplemental section (Supplementary Note 1; Supplementary Tables 1 and 2; Supplementary Fig. 3a). Whole exome sequencing (WES) from whole blood and variant analysis was performed by Ion AmplySeqTM Exome and Ion ProtonTM, with an average coverage of 85×, according to standard protocols (more information is available on request). The mRNASeq was performed on a muscle biopsy. Quality control, library preparation, sequencing platform and primary data analysis were performed as described elsewhere[41]. Analysis of the aligned data was done using DROP (Detection of RNAseq Outliers Pipeline)[42,43]. All data refer to GRCh37. Transcript ENST00000407025.5 was taken as reference for variant nomenclature.

## Statistics and reproducibility

No statistical method was used to predetermine sample size and the experiments were not randomized. Technical replicates are derived from independent experiments and represented as dots shown in the quantification histograms. For the quantification of live-cell imaging data, the number of selected cells was represented as both "n = x" in figure legends and individual dots. The results of statistical tests are indicated in the figure legends. The statistical significance of differences between two data sets was calculated using Student's unpaired two-tailed t-test. For the statistical significance of differences over two data sets, we conducted a non-parametric statistical analysis due to the violation of normality and homogeneity of variances assumptions required for ANOVA. Kruskal–Wallis H test was first used to compare median differences across all groups, which revealed significant differences. To identify specific group differences, we used the Mann–Whitney U test for pairwise comparisons due to its robustness in non-normal conditions. The P values are shown in the figures. All immunoblots are representative for at least three independent experiments, except for Figs. 5f and 6d, which only serve to demonstrate the expression of proteins. All the microscope pictures are representative for at least three independent experiments, except for Fig. 6h which is from two independent experiments.

## Reporting summary

Further information on research design is available in the Nature Portfolio Reporting Summary linked to this article.

## Data availability

The source data for all figures are available on Figshare (Figshare https://doi.org/10.6084/m9.figshare.25533646). The AlphaFold models of SDS22:PP1:I3 are available in ModelArchive (https://doi.org/10.5452/ma-tmp13). The RNA-sequencing data generated in this study were deposited in the GEO database: GSE249519. Raw MS data for this study are available at ProteomeXchange database: PXD050733. All other data can be found in the article, Supplementary Information or Figshare files.

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

## Acknowledgements

This project received financial support from the Research Foundation-Flanders (FWO Grant G090921N), the KU Leuven Research Fund (BOF Grant C14/20/101), and NIH/NIGMS R35GM119455. We are grateful to Dr. Dan Wu for performing flow-cytometry analysis shown in Fig. 6f. We thank Dr. Daniel Gerlich (Vienna Biocenter) for providing the HeLa Kyoto cells, Dr. Masato Kanemaki (School of Life Science, Sokendai) for the parental HCT116 Tet-OsTIR1 cell line, Dr. Hemmo Meyer (University of Duisburg-Essen) for the Strep-PP1γ HeLa cell line, and Dr. Stephen Taylor (University of Manchester) for the parental Flp-In T-REx HeLa cell line. Patient 1 was part of the URDCat Program (PERIS SLT002/16/00174). Finally, we would like to thank the patient and her family for their collaboration.

## Author contributions

X.C. obtained the data shown in Figs. 1c, g; 2d, h, i; 3c–f; 4c–e, g–k; 5f–l and 6a–g. M.L. generated the SDS22-degron cell line and obtained the data shown in Figs. 1b, d, j and 2e–g. G.v.d.H. generated the stable HeLa cell lines expressing SDS22 variants, and obtained the data shown in Figs. 1e, f, i; 3b and 6h. Z.C. generated the SDS22:PP1:I3 AlphaFold model (Fig. 4b, f), designed and validated the split-luciferase constructs (Fig. 5e –l), and obtained the data shown in Fig. 2a, b. J.d.P.G. acquired the data shown in Fig. 5a–c. S.L. prepared the samples for RNA sequencing and obtained the data shown in Fig. 1g, h. A.V.E. contributed to the design and analysis of the SDS-degron cell line, and the sample preparation for RNA seq and phosphoproteomics. S.K. provided expert advice on the performance and analyses of ITC experiments (Fig. 4k). E.G. and A.K. performed the phospho-proteome analysis (Fig. 2c). D.Nd., L.C.G., C.H.D., C.O., A.N. and R.U. obtained the clinical and genetic data shown in Fig. 3. M.B. and X.C. wrote the first version of the manuscript. M.B. coordinated the project. All authors read the final manuscript and agreed with its contents.

## Competing interests

The authors declare no competing interests
