## [Peer review file · Nature Communications]

REVIEWER COMMENTS

Reviewer #1 (Remarks to the Author):

The manuscript by Cao et al describes a series of well-designed experiments addressing important questions concerning the biogenesis of PP1 holoenzymes. The work was performed to a high standard, the text is exemplarily well-written and the data is presented in a clear manner and, for the most part, correctly interpreted to support the following conclusions/findings:

- i) SDS22 binding to nascent PP1 is required to stabilize the phosphatase and prevent its degradation/aggregation
- ii) SDS22 directly binds to I3, rendering its incorporation into a SDS-22:PP1:I3 holoenzyme irreversible.
- iii) p97/VCP specifically interacts with the ternary SDS-22:PP1:I3 holoenzyme, leading to extraction of both I3 and SDS22 and freeing (now) active PP1 catalytic subunit to bind other RIPPOs.
- iv) If PP1:SDS22 fails to interact with I3, inactive PP1:SDS22:canonical RIPPOs holoenzymes will form and compromise normal cellular function. Hence, co-extraction of I3 and SDS22 is essential to generate functional PP1 holoenzymes.

This is an elegant study that enhances greatly our molecular understanding of PP1 holoenzyme assembly, thus providing knowledge of significance to a broad readership. I do have few issues/comments, that, in my opinion, should be addressed to strengthen some of the conclusions and improve this already solid paper.

1 - Given the proposed model, one would also predict I3 as a RIPPOs required for the proliferation of cancer cell lines. However, that is not the case in the genome-wide RNAi-screens depicted in Fig S1e. Could the authors comment on this?

2- The authors claim that the prolonged mitosis of SDS22-depleted cells stems from a failure to silence the SAC. However, the authors should consider that the mitotic delay can also be attributed to unstable kinetochore-microtubule attachments that continuously signal for MCC assembly. Although the requirement of PP1 for efficient SAC silencing is well established, the role of PP1 and possibly SDS22 in promoting stable kinetochore-microtubule attachments remains debatable (PMID: 20921135; PMID: 34853300; PMID: 31808746). Therefore, to examine whether deficient SAC silencing accounts for the prolonged mitosis in SDS2-depleted cells, the metaphase duration should be quantified and evidence for stable kinetochore microtubule attachments (in metaphase

cells) should be provided. This can be done simply by staining for Astrin or by analyzing K-fibers upon Ca²⁺ or cold treatment.

3- It would be helpful to support the observed “strong staining for phosphorylated ERM proteins (pERM) in SDS22-depleted floating cells and cell clumps” with the corresponding quantifications. Since SDS22 depletion affected cell cycle progression, the authors may wish to compare pERM levels between “Untreated” and “Dox/IAA” cells that are in the same cell cycle stages: “Untreated” vs “Dox/IAA” in interphase; prometaphase/metaphase and anaphase.

4- An important conclusion of this work is that the PP1:I3 interaction is enhanced by SDS22. However, it has been recently shown that SDS22 weakens the affinity of I3 for PP1a in vitro via the I3 CCC motif (PMID: 38042495). In fact, from the His-I3-trap experiments presented in Fig 4, this seems to be the case with PP1 binding to I3 being reduced when in the presence of SDS22. It will be helpful to discuss these apparent contradictions.

5- The observation that the SDS22-binding mutant PP1-2KA was deficient in the binding of both SDS22 and I3 in the mClover-trapping experiments (Figure 5a), led the authors to conclude that efficient recruitment of I3 depends on SDS22. To strengthen this conclusion, and given the apparent discrepancies mentioned in the previous point, it would be advisable to perform the GST-Trap experiment depicted in Fig S5a with purified GST-PP1-2KA and confirm decreased binding of I3 to PP1-2KA.

6- The observation that the EGFP-I3-DE4A mutant fails to interact with SDS22, but binds as efficiently as EGFP-I3-WT to PP1 in the EGFP-pulldown experiments depicted in Fig 4h, does not fit with the proposed model of PP1:I3 interaction being enhanced by SDS22, and actually contrasts with what is shown in Fig 6a, where trapped EGFP-I3-DE4A interaction with PP1 is strongly compromised, or with the data presented in Fig 5i-k showing the irreversible nature of I3 binding to PP1:SDS22. A detailed and quantitative in vitro characterization (with purified components) of I3 binding to PP1 in the presence or absence of the I3:SDS22 interaction can help to clarify this point (as well as point 4). The authors may wish to consider comparing affinity measurements of PP1:I3-WT vs PP1:I3-DE4A in the presence of SDS22.

7- The lysate-based split-luciferase assays used to further inspect the role of SDS22 in the recruitment of I3 to PP1 are beautiful, elegant and informative. The results clearly demonstrate the irreversible nature of I3:PP1:SDS22 holoenzyme assembly driven by the simultaneous binding of I3 to PP1 and SDS22. The authors may wish to complement these nice piece of data with inhibitory kinetics of increasing concentrations of the RVxF peptide, particularly in the LgBiT-PP1-SDS22+I3-SmBiT reaction.

8- The observations that expression of SDS22-KR7A causes a mitotic delay which correlates with increased phosphorylation of pH3T3 and pERM are nice results that suggest the abnormal assembly of non-functional PP1:SDS22-KR7A:cRIPPO holoenzymes. However, to directly show that SDS22 mutated in its I3-inetaction site erroneously co-transfers with inactive PP1 to canonical RIPPOs, the levels of canonical RIPPOs that co-immunoprecipitate with SDS22-KR7A vs SDS22-WT should be evaluated.

9- p97/VCP specifically interacts with the ternary SDS22:PP1:I3 holoenzyme, and this is required to extract both I3 and SDS22 and thereby free PP1 to bind other RIPPOs. An important conclusion of this study is that the interaction between I3 and SDS22 enables the recruitment of p97/VCP and consequently the extraction of I3 and SDS22. Because SDS22 with a disabled I3-binding site (SDS22-KR7A) cannot be extracted from inactive PP1, it will lead to the formation of inactive PP1 holoenzymes. Would the expression of an SDS22-KR7A-I3 fusion restore the ability to produce active functional PP1 holoenzymes? Would it rescue the mitotic delay and mitotic hyperphosphorylation of PP1 substrates?

I hope the authors find these comments useful. This is a very good manuscript that provides important molecular insights into several aspects of PP1 biogenesis that had been elusive for decades. The discussion was a pleasure to read and presents a feasible model that reconciles a large body of data in the literature. I support the publication of this study in Nature Communications.

Reviewer #2 (Remarks to the Author):

PP1 is the most important phosphatase in eukaryotes. The function of the archaic and essential regulator of PP1, SDS22, has only recently started to emerge. The authors provide important new insight into the function of SDS22 that is vital for the understanding of how PP1 is regulated in the cell. They set up an elegant rapid degradation system to assess the relevance of SDS22 to proliferation and cell division. Moreover, they report on a patient with a mutation in SDS22 that underlines the relevance of SDS22 and is in line with the mechanistic data of the manuscript. Furthermore, the authors report a critical direct binding mechanism of SDS22 with a second partner of PP1, I3, that together form the ternary SDS22:PP1:I3 during PP1 biogenesis. They thus explain mechanistically why the formation of the complex is "irreversible" (requiring the AAA ATPase p97 for dissociation) and thus why it differs from functional PP1 holoenzyme complexes that are

dynamic. They also delineate the order of events showing that the initial partner of PP1 is SDS22 followed by I3 that is needed to recruit p97 and to induce SDS22:PP1 dissociation to allow holoenzyme formation.

Together, the reported findings are a timely major step in understanding the maturation of PP1 and therefore the basis for a myriad of signaling events of the cell. The data is generally high quality with innovative approaches. The manuscript still contains some unnecessary inconsistencies between the individual results and with published literature that the authors need to address.

(1) In Fig4c, the authors perform the only set of binding experiments with purified proteins. Generally, it would be desired to see more (and more sophisticated) biochemical approaches (SPR, BLI, ITC...) to confirm direct binding between SDS22 and I3 with purified proteins. Also, the authors need to validate in the biochemical assay the various mutants of the mutual binding sites described later in the manuscript (such as SBR mutants), which would also serve as excellent control for the pulldowns. As the data stand now in Fig4c, it seems as if SDS22 "fully" binds I3 even in the absence of PP1, and that PP1 may even compete for SDS22 with I3. However, this likely is an effect of the single set of chosen concentrations and does not really conform with the authors' conclusions. Using other biochemical techniques will allow the use of a range of concentrations and a more accurate assessment, consistent with the splitLuc assays.

(2) Related to the previous point: Generally, the authors conclude that the interfering with the SDS22:I3 binding has a fundamental effect on the stability and dynamics of the SDS22:PP1:I3 complex, which is supported by the splitLuc assays. However, in Fig. 4h-j, the authors present overexpression/pulldown data that show that the direct SDS22-I3 interaction is essential even for the formation of the ternary SDS22:PP1:I3 complex, which is difficult to reconcile with other data. These data seem to contradict the splitLuc results which show that the PP1-SDS22 fusion binds I3 better than PP1 alone. Moreover, the Peti lab recently published a SDS22:PP1 structure with a I3 fragment lacking the SBR (27-68) and presented SPR data showing that I3 and I3(27-68) have similar affinities for SDS22:PP1. Moreover, van den Boom et al (2021) showed that deleting residues 66-126 in I3 (including the SBR) does not decrease formation of the SDS22:PP1:I3 complex in cells. A full dependency on the SDS22:I3 interaction is also difficult to reconcile with the other binding sites of I3 for PP1 including the RVXF motif. The expectation would be that the SDS22:PP1:I3 still forms if the SDS22-I3 binding is compromised, but that this complex is more dynamic. The observed effects on binding of the respective other partner could be caused by exaggerated expression levels or long-term effects of the treatments. In any case, this issue needs to be solved because, again, this does not really fit with the conclusions.

(3) Fig. 5i-l: The splitLuc experiments are elegant and central to the conclusions. The approach should be done also for PP1 (without SDS22 fusion) comparing I3 and I3-SBR mutant. Moreover,

biological replica and statistics should be shown. Also, please explain why the scales are so different for the different conditions.

(4) Fig. 3 seems a bit thin which contrasts with the relevance. More data on the genetics could be of interest. Also, the effect of the mutation should be validated in model cells (stability of SDS22, PP1, effect on phospho-proteome,...). This could clarify whether the truncated SDS22 is expressed at lower rate, or destabilized and (for example) degraded by the proteasome.

(5) There is no section on data availability. The authors need to make the raw proteomics and sequencing data available to the reviewers, and to the public.

Minor:

The term "girl" in the abstract should be changed to "patient".

Why is the lower part of the blotting membrane cut in Fig 4J (strep trap)?

Fig. 5b: the cartoon is misleading: the reporter is linked to the C-terminus of I3 and not to an internal site.

Reviewer #3 (Remarks to the Author):

The manuscript by Cao et al. addresses the regulation of PP1, a critical phosphatase controlling many pivotal cellular pathways. Briefly, the authors generated a human cell line to acutely deplete SDS22, a negative regulator of PP1 that stabilizes its complex with I3. They combined this new cell line with classical, RNAi-based depletions. The PP1-SDS22-I3 complex traps PP1 in an inactive state before energy-dependent delivery to various proteins harboring specific PP1 docking motifs (SLIMs). The exact function of SDS22 in the stabilization of the PP1-I3-SDS22 complex is being debated. While not exactly acute, as it required several hours, depletion of SDS22 through AID-

mediated degron variously affected the cell cycle, e.g. delaying G1-S progression and causing a temporary arrest in M-phase. These conditions correlated with changes in expression and phosphorylation patterns of various proteins, especially cell cycle regulators. Using AI-based prediction methods, the authors identified a previously unrecognized ionic interaction of SDS22 with I3. After characterizing the interaction mechanism and identifying appropriate mutants, the authors probed the relevance of the SDS22-I3 interaction for the mechanism of PP1 transfer to SLIMs motifs of physiological PP1 binding partners. Probably the most crucial observation is that the interaction of I3 with SDS22 is essential to stabilize the interaction of I3 with PP1. This was shown to have very important implications for the mechanism of activation of PP1. I3 has been shown to mediate an interaction with the p97/VCP ATPase that frees PP1 from it, licensing its interaction with other PP1-binding proteins. The authors now show that I3 is also required to free PP1 from its SDS22 inhibitor. Disruption of the SDS22-I3 binding site was shown to be compatible with the interaction of PP1 with other PP1 binding partners, but the resulting complexes remained bound to the SDS22 inhibitor (whose binding affinity for PP1 is extremely high). Thus, collectively, these new data identify I3 as the protein required to remove SDS22 from PP1, which, as far as I can tell, is a novel and very important conclusion.

This is a lovely manuscript that I am happy to recommend for publication. The technical quality of the work is outstanding. The presentation is very clear (a note on this below). The conclusions are novel and of great mechanistic significance for the field, and the Discussion does a very good job at presenting them concisely and, at least in my view, in a balanced manner. The manuscript reports new, original observations that provide a much deeper conceptual perspective on the mechanism of PP1 activation, as well as new tools that will be of great significance for further research.

Minor points:

Line 80: Question mark missing

Line 379: There is no panel 2i, or rather there is one but it lacks the 'i' label

In my view, the most exciting and novel observations are those on the I3-SDS22 interaction presented in Figures 4-6. The manuscript would have a stronger start if these figures could be presented first. Inverting the narrative, thus discussing Figures 1-3 (which focus on SDS22) after Figures 4-6 may be possible if one considers that Figures 4-6 raise the question why SDS22 is at all needed. Figures 1-2 provide an at least partial answer to this question (stabilization of PP1) and Figure 3 further demonstrates the relevance of this protein. This is only a suggestion for the authors.

Reply to the comments of the reviewers

We thank the reviewers for their constructive and detailed review reports, which have helped us to prepare a much-improved manuscript. As requested and detailed below, we have performed numerous additional experiments and introduced textual changes. All changes in the manuscript are indicated in red.

Reviewer #1

The manuscript by Cao et al describes a series of well-designed experiments addressing important questions concerning the biogenesis of PP1 holoenzymes. The work was performed to a high standard, the text is exemplarily well-written and the data is presented in a clear manner and, for the most part, correctly interpreted to support the following conclusions/findings:

- i) SDS22 binding to nascent PP1 is required to stabilize the phosphatase and prevent its degradation/aggregation
- ii) SDS22 directly binds to I3, rendering its incorporation into a SDS-22:PP1:I3 holoenzyme irreversible.
- iii) p97/VCP specifically interacts with the ternary SDS-22:PP1:I3 holoenzyme, leading to extraction of both I3 and SDS22 and freeing (now) active PP1 catalytic subunit to bind other RIPPOs.
- iv) If PP1:SDS22 fails to interact with I3, inactive PP1:SDS22:canonical RIPPOs holoenzymes will form and compromise normal cellular function. Hence, co-extraction of I3 and SDS22 is essential to generate functional PP1 holoenzymes.

This is an elegant study that enhances greatly our molecular understanding of PP1 holoenzyme assembly, thus providing knowledge of significance to a broad readership. I do have few issues/comments, that, in my opinion, should be addressed to strengthen some of the conclusions and improve this already solid paper.

Reply

We were happy to read that the reviewer found our manuscript solid, significant and well-written.

1 - Given the proposed model, one would also predict I3 as a RIPPOs required for the proliferation of cancer cell lines. However, that is not the case in the genome-wide RNAi-screens depicted in Fig S1e. Could the authors comment on this?

Reply

SDS22 is required for both PP1 stabilisation and holoenzyme assembly, while I3 is only required for holoenzyme assembly with newly translated PP1. Upon SDS22 depletion, newly translated PP1 is unstable (Fig. 5b), resulting in a loss of PP1 (Figs. 2h, 2i) and a severely decreased cell proliferation (Fig. 1c and Fig. S1e). I3 depletion does not affect the stability of newly translated PP1 (Fig. 5c), which may even be used for holoenzyme assembly in a p97/VCP-independent manner, as its RVxF-docking site is accessible for canonical RIPPOs in the absence of (irreversibly-bound) I3. This may explain why a short-term, siRNA-mediated depletion of I3 has a less severe proliferation phenotype than SDS22 depletion (Figure S1e). However, a long-term depletion of I3 also results in a proliferation-defect

(<https://depmap.org/portal/>), which likely stems from a gradual loss of (functional) PP1 holoenzymes. We have discussed this notion in the revised manuscript (lines 452-459).

2- The authors claim that the prolonged mitosis of SDS22-depleted cells stems from a failure to silence the SAC. However, the authors should consider that the mitotic delay can also be attributed to unstable kinetochore-microtubule attachments that continuously signal for MCC assembly. Although the requirement of PP1 for efficient SAC silencing is well established, the role of PP1 and possibly SDS22 in promoting stable kinetochore-microtubule attachments remains debatable (PMID: 20921135; PMID: 34853300; PMID: 31808746). Therefore, to examine whether deficient SAC silencing accounts for the prolonged mitosis in SDS22-depleted cells, the metaphase duration should be quantified and evidence for stable kinetochore microtubule attachments (in metaphase cells) should be provided. This can be done simply by staining for Astrin or by analyzing K-fibers upon Ca²⁺ or cold treatment.

Reply

As requested, we have performed time-lapse imaging to address this point. In accordance with our conclusions, the duration of metaphase was prolonged in SDS22-depleted cells (new Figs. S2b and S2c). However, SDS22 depletion had no effect on the prophase → metaphase duration (new Fig. S2d). These data (discussed lines 156-160) strongly argue against a rate-limiting effect of SDS22 depletion on achieving stable kinetochore-microtubule attachments and are consistent with the proposed (indirect) contribution of SDS22 to SAC silencing.

3- It would be helpful to support the observed “strong staining for phosphorylated ERM proteins (pERM) in SDS22-depleted floating cells and cell clumps” with the corresponding quantifications. Since SDS22 depletion affected cell cycle progression, the authors may wish to compare pERM levels between “Untreated” and “Dox/IAA” cells that are in the same cell cycle stages: “Untreated” vs “Dox/IAA” in interphase; prometaphase/metaphase and anaphase.

Reply

We have quantified the stainings for pERM, as requested (new Figs. 2F and S2h). In addition, we have performed immunoblots for pERM in synchronized cells (nocodazole arrest and washout), confirming ERM hyperphosphorylation in synchronized SDS22-depleted cells (new Fig. S2i).

4- An important conclusion of this work is that the PP1:I3 interaction is enhanced by SDS22. However, it has been recently shown that SDS22 weakens the affinity of I3 for PP1a in vitro via the I3 CCC motif (PMID: 38042495). In fact, from the His-I3-trap experiments presented in Fig 4, this seems to be the case with PP1 binding to I3 being reduced when in the presence of SDS22. It will be helpful to discuss these apparent contradictions.

Reply

At first glance, the His-I3 trap experiment in Fig. 4c indeed suggests that I3 binds somewhat less well to PP1 in the presence of SDS22. However, we performed this experiment 3 times and a decreased binding was only detected once (see pictures below). For that reason, we replaced Fig. 4c with a more representative experiment (exp. 1). We also want to point out that these experiments were performed at high, saturating concentrations of PP1 (0.3 μM) and I3 (2 μM), as is now indicated in the figure legend, which likely precludes the detection of a stimulatory effect of SDS22 on the PP1:I3 interaction. Indeed, I3 by itself already binds with high affinity to PP1, and its irreversible binding in the presence of SDS22 only matters in a cellular context to prevent out-competition by canonical RIPPOs. Overexpression of I3 (out-competition of other RIPPOs for PP1 binding) has a similar effect as SDS22 binding (excluding competition

by other RIPPOs due to irreversible binding of I3), and diminishes the contribution of the SDS22:I3 interaction to the recruitment of I3.

Fig. R1. Three repeats of Figure 4c. Exps 1 and 2 are immunoblots, Exp 3 is a Coomassie staining

It has indeed been reported recently that SDS22 weakens the affinity of I3 for PP1 α *in vitro* (PMID: 38042495). However, the latter studies were performed with N-terminally truncated proteins [PP1-(7-320); SDS22-(56-360)], which may affect their interaction with I3, as is now discussed (p 13, line 17-21).

5- The observation that the SDS22-binding mutant PP1-2KA was deficient in the binding of both SDS22 and I3 in the mClover-trapping experiments (Figure 5a), led the authors to conclude that efficient recruitment of I3 depends on SDS22. To strengthen this conclusion, and given the apparent discrepancies mentioned in the previous point, it would be advisable to perform the GST-Trap experiment depicted in Fig S5a with purified GST-PP1-2KA and confirm decreased binding of I3 to PP1-2KA.

Reply

We have performed (a variant of) the experiment shown in Fig. S5a, using purified GST-PP1-WT and GST-PP1-2KA (new Fig. S5b). The data confirmed that GST-PP1-WT bound to SDS22 and I3, while GST-PP1-2KA only bound to I3. Importantly, the GST-PP1-2KA:I3 complex was completely disrupted by the addition of a 10-fold molar excess of NIPP1-(143-224), but the GST-PP1-WT:I3 complex was only partially dissociated by NIPP1-(145-224). These data provide additional, independent evidence for the stabilisation of PP1:I3 by SDS22.

6- The observation that the EGFP-I3-DE4A mutant fails to interact with SDS22, but binds as efficiently as EGFP-I3-WT to PP1 in the EGFP-pulldown experiments depicted in Fig 4h, does not fit with the proposed model of PP1:I3 interaction being enhanced by SDS22, and actually contrasts with what is shown in Fig 6a, where trapped EGFP-I3-DE4A interaction with PP1 is

strongly compromised, or with the data presented in Fig 5i-k showing the irreversible nature of I3 binding to PP1:SDS22. A detailed and quantitative in vitro characterization (with purified components) of I3 binding to PP1 in the presence or absence of the I3:SDS22 interaction can help to clarify this point (as well as point 4). The authors may wish to consider comparing affinity measurements of EGFP- I3-WT and EGFP-I3-DE4A bind equally well to PP1 in pull down experiments (in the presence of SDS22).

Reply

In our reply to comment 5 of reviewer 1, we already discussed novel data, using purified components, demonstrating that the PP1:I3 interaction is enhanced by SDS22 (new Fig. S5b).

The apparent contradiction with the data in Fig. 4c was already addressed above (see reply to comment 4 of reviewer 1). As for the binding of PP1 to ectopically expressed I3, we detected less PP1 in traps of EGFP-I3-DE4A as compared to traps of EGFP-I3-WT (decrease of $35 \pm 10\%$, $n=6$), as is now specified in the text (p 9, line 21). These differences are small because overexpression of I3 (out-competition of canonical RIPPOs) has a similar effect on PP1 binding as the binding of SDS22 to I3 (PP1:I3 interaction becomes irreversible, precluding competition with canonical RIPPOs). Hence, the stimulatory effect of SDS22 on the PP1:I3 interaction is alleviated by I3 overexpression (see also reply to point 4 of reviewer 1).

7- The lysate-based split-luciferase assays used to further inspect the role of SDS22 in the recruitment of I3 to PP1 are beautiful, elegant and informative. The results clearly demonstrate the irreversible nature of I3:PP1:SDS22 holoenzyme assembly driven by the simultaneous binding of I3 to PP1 and SDS22. The authors may wish to complement these nice piece of data with inhibitory kinetics of increasing concentrations of the RVxF peptide, particularly in the LgBiT-PP1-SDS22+I3-SmBiT reaction.

Reply

We have repeated all split-luciferase assays shown in Figs. 5i-k with various concentrations of RVxF peptide (0-50 μM), as requested (new Fig. S5c). The data confirm and extend the data of Fig. 5i-k. Since these figure panels are a bit overloaded, we have chosen to show them in the supplemental section.

8- The observations that expression of SDS22-KR7A causes a mitotic delay which correlates with increased phosphorylation of pH3T3 and pERM are nice results that suggest the abnormal assembly of non-functional PP1:SDS22-KR7A:cRIPPO holoenzymes. However, to directly show that SDS22 mutated in its I3-ination site erroneously co-transfers with inactive PP1 to canonical RIPPOs, the levels of canonical RIPPOs that co-immunoprecipitate with SDS22-KR7A vs SDS22-WT should be evaluated.

Reply

The reviewer may have missed it, but Fig. 6b does show that SDS22-KR7A, in contrast to SDS22-WT, is present in complexes with canonical RIPPOs such as MYPT1 and RepoMan.

9- p97/VCP specifically interacts with the ternary SDS22:PP1:I3 holoenzyme, and this is required to extract both I3 and SDS22 and thereby free PP1 to bind other RIPPOs. An important conclusion of this study is that the interaction between I3 and SDS22 enables the recruitment of p97/VCP and consequently the extraction of I3 and SDS22. Because SDS22 with a disabled I3-binding site (SDS22-KR7A) cannot be extracted from inactive PP1, it will lead to the formation of inactive PP1 holoenzymes. Would the expression of an SDS22-KR7A-I3 fusion

restore the ability to produce active functional PP1 holoenzymes? Would it rescue the mitotic delay and mitotic hyperphosphorylation of PP1 substrates?

Reply

This is an interesting suggestion but a rescue with a SDS22-KR7A-I3 fusion is not as straightforward as it looks at first glance. Indeed, overexpression of any RIPPO with an RVxF motif, including the SDS22-KR7A-I3 fusion, will competitively disrupt other PP1 holoenzymes, including holoenzymes that are needed for mitotic progression (doi: 10.1242/jcs.175588). For this reason, the overexpression of SDS22-KR7A-I3, I3 or any other RVxF-based RIPPO will cause a mitotic arrest and can therefore not be used for mitotic 'rescue' experiments.

I hope the authors find these comments useful. This is a very good manuscript that provides important molecular insights into several aspects of PP1 biogenesis that had been elusive for decades. The discussion was a pleasure to read and presents a feasible model that reconciles a large body of data in the literature. I support the publication of this study in Nature Communications.

Reply

We thank the reviewer for her/his constructive and highly relevant comments.

Reviewer #2 (Remarks to the Author):

PP1 is the most important phosphatase in eukaryotes. The function of the archaic and essential regulator of PP1, SDS22, has only recently started to emerge. The authors provide important new insight into the function of SDS22 that is vital for the understanding of how PP1 is regulated in the cell. They set up an elegant rapid degradation system to assess the relevance of SDS22 to proliferation and cell division. Moreover, they report on a patient with a mutation in SDS22 that underlines the relevance of SDS22 and is in line with the mechanistic data of the manuscript. Furthermore, the authors report a critical direct binding mechanism of SDS22 with a second partner of PP1, I3, that together form the ternary SDS22:PP1:I3 during PP1 biogenesis. They thus explain mechanistically why the formation of the complex is "irreversible" (requiring the AAA ATPase p97 for dissociation) and thus why it differs from functional PP1 holoenzyme complexes that are dynamic. They also delineate the order of events showing that the initial partner of PP1 is SDS22 followed by I3 that is needed to recruit p97 and to induce SDS22:PP1 dissociation to allow holoenzyme formation.

Together, the reported findings are a timely major step in understanding the maturation of PP1 and therefore the basis for a myriad of signaling events of the cell. The data is generally high quality with innovative approaches. The manuscript still contains some unnecessary inconsistencies between the individual results and with published literature that the authors need to address.

Reply

We were flattered to read that the reviewer found our work innovative, of high quality and a major advance in understanding PP1 maturation.

(1) In Fig4c, the authors perform the only set of binding experiments with purified proteins. Generally, it would be desired to see more (and more sophisticated) biochemical approaches (SPR, BLI, ITC...) to confirm direct binding between SDS22 and I3 with purified proteins. Also, the authors need to validate in the biochemical assay the various mutants of the mutual binding sites described later in the manuscript (such as SBR mutants), which would also serve

as excellent control for the pulldowns. As the data stand now in Fig4c, it seems as if SDS22 "fully" binds I3 even in the absence of PP1, and that PP1 may even compete for SDS22 with I3. However, this likely is an effect of the single set of chosen concentrations and does not really conform with the authors' conclusions. Using other biochemical techniques will allow the use of a range of concentrations and a more accurate assessment, consistent with the splitLuc assays.

Reply

We also performed binding-competition studies with purified components in Fig. S5a and the new Fig. S5b, requested by reviewer 1. In addition, we have now included ITC interaction assays with purified SDS22-WT, SDS22-7KRA and I3-WT (new Fig. 4k and S4g). The data confirm that SDS22 binds directly to I3 ($K_d = 7.6 \pm 5.5 \mu\text{M}$) and that this binding is strongly reduced by mutation of the I3-binding site of SDS22 ($K_d = 127.6 \pm 28.1 \mu\text{M}$).

As already indicated (see reply to comment 4 of reviewer 1), the His-I3 trap experiment shown in Fig. 4c was not really representative for all the repeat experiments that were performed. For that reason, we replaced Fig. 4c with a more representative experiment. We also want to point out that these experiments were performed at high saturating concentrations of PP1 (0.3 μM) and I3 (2 μM), as is now indicated in the figure legend, which likely precludes the detection of a stimulatory effect of SDS22 on the PP1:I3 interaction. Indeed, I3 by itself already binds with high affinity to PP1, and its irreversible binding in the presence of SDS22 only matters in a cellular context to prevent out-competition by canonical RIPPOs. Overexpression of I3 (out-competition of other RIPPOs for PP1 binding) has a similar effect as SDS22 binding (excluding competition by other RIPPOs due to irreversible binding of I3). Hence, the stimulatory effect of SDS22 on the PP1:I3 interaction is alleviated by I3 overexpression.

(2) Related to the previous point: Generally, the authors conclude that the interfering with the SDS22:I3 binding has a fundamental effect on the stability and dynamics of the SDS22:PP1:I3 complex, which is supported by the splitLuc assays. However, in Fig. 4h-j, the authors present overexpression/pulldown data that show that the direct SDS22-I3 interaction is essential even for the formation of the ternary SDS22:PP1:I3 complex, which is difficult to reconcile with other data. These data seem to contradict the splitLuc results which show that the PP1-SDS22 fusion binds I3 better than PP1 alone. Moreover, the Peti lab recently published a SDS22:PP1 structure with a I3 fragment lacking the SBR (27-68) and presented SPR data showing that I3 and I3(27-68) have similar affinities for SDS22:PP1. Moreover, van den Boom et al (2021) showed that deleting residues 66-126 in I3 (including the SBR) does not decrease formation of the SDS22:PP1:I3 complex in cells. A full dependency on the SDS22:I3 interaction is also difficult to reconcile with the other binding sites of I3 for PP1 including the RVXF motif. The expectation would be that the SDS22:PP1:I3 still forms if the SDS22-I3 binding is compromised, but that this complex is more dynamic. The observed effects on binding of the respective other partner could be caused by exaggerated expression levels or long-term effects of the treatments. In any case, this issue needs to be solved because, again, this does not really fit with the conclusions.

Reply

It is correct that SDS22 is not needed to form a PP1:I3 complex and that a stable ternary SDS22:PP1:I3 complex is formed, even when the SDS22-binding domain of I3 is deleted. In fact, I3 has a binding affinity for PP1 that is similar to that of other RIPPOs. However, the key point of our findings is that SDS22 makes the recruitment of I3 irreversible (Figs. 5i-k), which prevents out-competition of I3 by other RIPPOs. This mechanism ensures that newly translated

PP1 selectively interacts with SDS22 and I3, until PP1 is transferred by p97/VCP to canonical RIPPOs.

It has indeed been reported recently that SDS22 weakens the affinity of I3 for PP1 α *in vitro* (PMID: 38042495). However, the latter studies were performed with N-terminally nicked proteins [PP1-(7-320); SDS22-(56-360)], which may affect their interaction with I3, as is now discussed (p 13, line 23-27).

As for Figs. 4h and i, these are experiments where EGFP-tagged I3 or SDS22 variants are massively overexpressed. Fig. 4h shows that I3 still binds to PP1 when its SDS22-interacted site is mutated, consistent with the high-affinity binding of I3 to PP1 (see first para to this reply). Given their massive overexpression, it is not surprising that EGFP-I3 and EGFP-I3-DE4A bind nearly equally well to PP1 in co-immunoprecipitation experiments (see reply to comment 6 of reviewer 1 for details), if they are present at concentrations above the K_d for their interaction with PP1. Fig. 4i shows that SDS22 still binds to PP1 in the absence of I3, even more because SDS22 is no longer extracted before transfer of PP1 to canonical RIPPOs. These data do not contradict with our conclusion that SDS22 makes the binding of newly translated PP1 to I3 selective and irreversible.

(3) Fig. 5i-l: The splitLuc experiments are elegant and central to the conclusions. The approach should be done also for PP1 (without SDS22 fusion) comparing I3 and I3-SBR mutant. Moreover, biological replica and statistics should be shown. Also, please explain why the scales are so different for the different conditions.

Reply

We have now also performed kinetic split-luciferase experiments with LgBiT-PP1 and either I3-WT-SmBiT or I3-DE4A-SmBiT, as requested (new Fig S5d). The data confirm that both sensors are competitively disrupted with an RVxF-peptide. In fact, lower concentrations of RVxF peptide are needed to disrupt PP1:I3-DE4A in cell lysates (new Fig. S5d) than are needed to disrupt PP1:I3-WT (Fig. S5c), consistent with a stabilizing effect of SDS22 binding on the I3:PP1 interaction.

The experiments shown in Figs. 5i-l were performed 3 times, with newly prepared lysates. Moreover, based on comment 7 of reviewer 1, we have repeated the experiments shown in Fig. 5i-k an additional 2 times, using different concentrations of competitor (new figure S5c), with an identical outcome. The number of repeats of the kinetic split luciferase experiments is now specified in the Figure legends. We did not find it appropriate to show averages \pm SD for these kinetic experiments (continuous reading), in contrast to split-luciferase assays with only end-point reading (Figs. 5g-h).

The scales in Figs. 5i-l are very different because different sensors are used. The signal that is generated by a sensor is determined by (1) the extent of interaction between LgBiT-POI1 and SmBiT-POI2, and (2) the efficiency of complementation between LgBiT and SmBiT in the LgBiT-POI1:SmBiT-POI2 complex, which is determined by the relative positions of LgBit and SmBiT with respect to each other. To avoid confusion, we have now presented the sensor activities plotted as a percentage of the signal just before addition of competitor ($t = 12$ min).

(4) Fig. 3 seems a bit thin which contrasts with the relevance. More data on the genetics could be of interest. Also, the effect of the mutation should be validated in model cells (stability of SDS22, PP1, effect on phospho-proteome,...). This could clarify whether the truncated SDS22 is expressed at lower rate, or destabilized and (for example) degraded by the proteasome.

Reply

We agree that Fig. 3 in the first version of the manuscript was a bit thin. As requested, we have compared the fate of EGFP-tagged SDS22-WT and SDS22-W302* (the SDS22 mutant of patient P1) in transiently transfected HEK293T cells. Unlike SDS22-WT, SDS22-W302* did not associate with PP1, I3 or BCLAF1 (new Fig. 3c), showing that the mutant is dysfunctional with respect to ligand binding. Moreover, EGFP-tagged EGFP-SDS22-W302* was expressed at much lower levels than EGFP-SDS22-WT (new Figs. 3d and 3e), but this phenotype was largely rescued by addition of the proteasome inhibitor MG132. This suggested that SDS22-W302* is less abundant because of an increased degradation rate. Finally, we noted that SDS22-W302*, before and after treatment with MG132, largely appeared in granules, hinting at an increased tendency to aggregate (new Fig. 3f). Together, these data show that the mutant version of SDS22 in patient P1 (SDS22-W302*) is dysfunctional, less soluble and rapidly targeted for proteasomal degradation. These data validate and extend our observations on SDS22-W302* in fibroblasts of P1, and demonstrated that this mutant is deficient in ligand binding, less soluble and rapidly targeted for proteasomal degradation.

We have now included all the clinical data sets of patient P1, as also requested by the editorial office, as a supplemental text file (Text file T1) to Fig. 3.

(5) There is no section on data availability. The authors need to make the raw proteomics and sequencing data available to the reviewers, and to the public.

Reply

We have now included a section on data availability (lines 713-722). All primary datasets, including raw proteomics, sequencing data and AlphaFold models will be publicly accessible.

Minor:

The term "girl" in the abstract should be changed to "patient".

Reply

The text has been changed as requested.

Why is the lower part of the blotting membrane cut in Fig 4J (strep trap)?

Reply

The EGFP-blot in Fig. 4j do not show masses below 30 kDa because the lower part of the blot was used for Strep-I3 detection (ca. 25 kDa; upper panel). This is now indicated in the legend. Moreover, all uncropped blots shown in the manuscript are available on FigShare.

Fig. 5b: the cartoon is misleading: the reporter is linked to the C-terminus of I3 and not to an internal site.

Reply

We presume that reviewer refers to Fig. 5d, not Fig. 5b. The cartoon has been corrected as suggested.

Reviewer #3

The manuscript by Cao et al. addresses the regulation of PP1, a critical phosphatase controlling many pivotal cellular pathways. Briefly, the authors generated a human cell line to acutely deplete SDS22, a negative regulator of PP1 that stabilizes its complex with I3. They combined this new cell line with classical, RNAi-based depletions. The PP1-SDS22-I3 complex traps PP1

in an inactive state before energy-dependent delivery to various proteins harboring specific PP1 docking motifs (SLIMs). The exact function of SDS22 in the stabilization of the PP1-I3-SDS22 complex is being debated. While not exactly acute, as it required several hours, depletion of SDS22 through AID-mediated degron variously affected the cell cycle, e.g. delaying G1-S progression and causing a temporary arrest in M-phase. These conditions correlated with changes in expression and phosphorylation patterns of various proteins, especially cell cycle regulators. Using AI-based prediction methods, the authors identified a previously unrecognized ionic interaction of SDS22 with I3. After characterizing the interaction mechanism and identifying appropriate mutants, the authors probed the relevance of the SDS22-I3 interaction for the mechanism of PP1 transfer to SLIMs motifs of physiological PP1 binding partners. Probably the most crucial observation is that the interaction of I3 with SDS22 is essential to stabilize the interaction of I3 with PP1. This was shown to have very important implications for the mechanism of activation of PP1. I3 has been shown to mediate an interaction with the p97/VCP ATPase that frees PP1 from it, licensing its interaction with other PP1-binding proteins. The authors now show that I3 is also required to free PP1 from its SDS22 inhibitor. Disruption of the SDS22-I3 binding site was shown to be compatible with the interaction of PP1 with other PP1 binding partners, but the resulting complexes remained bound to the SDS22 inhibitor (whose binding affinity for PP1 is extremely high). Thus, collectively, these new data identify I3 as the protein required to remove SDS22 from PP1, which, as far as I can tell, is a novel and very important conclusion.

This is a lovely manuscript that I am happy to recommend for publication. The technical quality of the work is outstanding. The presentation is very clear (a note on this below). The conclusions are novel and of great mechanistic significance for the field, and the Discussion does a very good job at presenting them concisely and, at least in my view, in a balanced manner. The manuscript reports new, original observations that provide a much deeper conceptual perspective on the mechanism of PP1 activation, as well as new tools that will be of great significance for further research.

Reply

We were thrilled to read that the reviewer found our work of outstanding technical quality and of great mechanistic significance.

Minor points:

Line 80: Question mark missing

Reply

The question mark has been added.

Line 379: There is no panel 2i, or rather there is one but it lacks the 'i' label.

Reply

Panel 2i is now properly labeled.

In my view, the most exciting and novel observations are those on the I3-SDS22 interaction presented in Figures 4-6. The manuscript would have a stronger start if these figures could be presented first. Inverting the narrative, thus discussing Figures 1-3 (which focus on SDS22) after Figures 4-6 may be possible if one considers that Figures 4-6 raise the question why SDS22 is at all needed. Figures 1-2 provide an at least partial answer to this question

(stabilization of PP1) and Figure 3 further demonstrates the relevance of this protein. This is only a suggestion for the authors.

Reply

We have given this suggestion a lot of consideration but, eventually, choose not to change the narrative of the story. The major reason for this choice is that the presented story line is in accordance with the way the project has actually evolved. We first found that a depletion of SDS22 in the degon cell line results in the destabilisation of PP1, the best characterized interactor of SDS22 (Figs. 1-3). It is only afterwards that we discovered that SDS22 is also important for the function of I3, which we identified as a novel direct interactor of SDS22 (Figs. 3-6).

REVIEWERS' COMMENTS

Reviewer #1 (Remarks to the Author):

In this revised manuscript, Cao et al. have addressed all my previous concerns. The study has been further strengthened, building upon what was already a solid foundation. It provides valuable insights that contribute to our understanding of PP1 biogenesis

Reviewer #2 (Remarks to the Author):

In an impressive effort, the authors addressed all of this reviewer's concerns. In particular, they reveal the cellular defect of the SDS22 disease mutant protein. Moreover, they added a whole set of new interaction data that further support the proposed model for SDS22 and I3 binding to PP1. These data also convincingly clarify a misconception of a previous paper that I3 would weaken the SDS22-PP1 interaction. Overall, the data conclusively explains the function of SDS22 during PP1 biogenesis and reveal how I3 (and its physical and functional interaction with SDS22) renders this binding "irreversible", and the role of I3 in SDS22-PP1-I3 disassembly by p97-p37. I highly recommend publication.

Reviewer #3 (Remarks to the Author):

I thank the authors for their thorough revision of an already excellent manuscript. The manuscript can be published in its present form, but I have a slight concern regarding the representation and significance of the new ITC data in the panel 4k. If I understand these panels correctly, the reaction is endothermic (positive kcal values), which implies that it is driven by entropy changes. In addition, there does not seem to be a clear binding titration. To me, this looks more like a de-aggregation phenomenon upon dilution. The authors are highly conscientious and I am sure that they will consider this point careful before making a decision whether to display the data or give the matter further consideration.

Reply to the remaining comment of 1 reviewer

We were pleased to read that Reviewers 1 and 2 did not formulate additional concerns. The single remaining comment of Reviewer 3 is addressed below.

Reviewer #1

In this revised manuscript, Cao et al. have addressed all my previous concerns. The study has been further strengthened, building upon what was already a solid foundation. It provides valuable insights that contribute to our understanding of PP1 biogenesis

Reply:

No remaining comments to address.

Reviewer #2

In an impressive effort, the authors addressed all of this reviewer's concerns. In particular, they reveal the cellular defect of the SDS22 disease mutant protein. Moreover, they added a whole set of new interaction data that further support the proposed model for SDS22 and I3 binding to PP1. These data also convincingly clarify a misconception of a previous paper that I3 would weaken the SDS22-PP1 interaction. Overall, the data conclusively explains the function of SDS22 during PP1 biogenesis and reveal how I3 (and its physical and functional interaction with SDS22) renders this binding "irreversible", and the role of I3 in SDS22-PP1-I3 disassembly by p97-p37. I highly recommend publication.

Reply:

No remaining comments to address.

Reviewer #3

I thank the authors for their thorough revision of an already excellent manuscript. The manuscript can be published in its present form, but I have a slight concern regarding the representation and significance of the new ITC data in the panel 4k. If I understand these panels correctly, the reaction is endothermic (positive kcal values), which implies that it is driven by entropy changes. In addition, there does not seem to be a clear binding titration. To me, this looks more like a de-aggregation phenomenon upon dilution. The authors are highly conscientious and I am sure that they will consider this point careful before making a decision whether to display the data or give the matter further consideration.

Reply:

The binding isotherms presented in Figure 4k represent the residual heat following subtraction of the control ‘heat of dilution’ experiment (empty cell vs injectant in same concentration and conditions used for the binding experiment) from the binding experiment. Hence, the binding isotherms represent true binding curves, not a ‘heat of dilution’ effect. We have now added a sentence in the Method section under ITC to clarify this point: ‘The heat of dilution for the injectant, under the same conditions, was subtracted from the binding experiment, before analysis of the binding’.

The reviewer is correct in stating that there is no clear binding titration. To achieve saturation one would need to use higher concentrations of injectant, which was not feasible due to aggregation of the protein. We performed the ITC experiments using a concentration of the injectant (50 μM His-I3) that does not cause aggregation. This enabled us to generate a real binding isotherm, but came with the penalty of not allowing saturation of the binding phenomenon. We have added a sentence in the Methods section under ITC (line 651) to indicate this limitation: ‘The concentration of injectant that could be used (50 μM His-I3) was limited by aggregation at higher concentrations, accounting for the non-saturation of the binding curves.’